# Sensitivity model study of regional mercury dispersion in the atmosphere

Christian N. Gencarelli[1], Johannes Bieser[2,3], Francesco Carbone[1], Francesco De Simone[1], Ian M. Hedgecock[1], Volker Matthias[2], Oleg Travnikov[4], Xin Yang[5], and Nicola Pirrone[6]

[1]CNR-Institute of Atmospheric Pollution Research, Division of Rende, Rende, Italy
[2]Institute of Coastal Research, Helmholtz-Zentrum Geesthacht, Geesthacht, Germany
[3]National Aeronautics and Space Center (DLR), Oberpfaffenhofen, Weßling, Germany
[4]Meteorological Synthesizing Centre, East of EMEP, 2nd Roshchinsky proezd, 8/5, 115419 Moscow, Russia
[5]British Antarctic Survey,Cambridge,United Kingdom
[6]CNR-Institute of Atmospheric Pollution Research, Monterotondo, Rome, Italy

*Correspondence to:* C. N. Gencarelli (christian.gencarelli@iia.cnr.it)

**Abstract.** Atmospheric deposition is the most important pathway by which Hg reaches marine ecosystems, where it can be methylated and enter the base of food chain. The deposition, the transport and chemical interactions of atmospheric Hg has been simulated over Europe for the year 2013 in the framework of the Global Mercury Observation System (GMOS) project, performing 14 different model sensitivity tests using two high resolution three-dimensional Chemical Transport Models (CTMs), varying the anthropogenic emissions data sets, atmospheric Br input fields, the Hg oxidation schemes and the modelling domain boundary condition input. Sensitivity simulation results were compared with observations from 28 monitoring sites in Europe, to assess model performance and particularly to analyse the influence of anthropogenic emission speciation and the $Hg^0_{(g)}$ atmospheric oxidation mechanism. The contribution of anthropogenic Hg emissions, their speciation and vertical distribution is crucial to the simulated concentration and deposition fields, as is also the choice of $Hg^0_{(g)}$ oxidation pathway. The areas most sensitive to changes in Hg emission speciation and the emission vertical distribution are those near major sources, but also the Aegean and the Black Seas, the English Channel, the Skagerrak Strait and the North German coast. Considerable influence was found also evident over the Mediterranean, the North and Baltic Sea, some influence is seen over continental Europe, while this difference is least over the north-western part of the modelling domain, which includes the Norwegian Sea and Iceland. The Br oxidation pathway produces more $Hg^{II}_{(g)}$ in the lower model levels, but overall wet deposition is lower in comparison to the simulations which employ an $O_3$/OH oxidation mechanism. The necessity to perform continuous measurements of speciated Hg, to investigate the local impacts of Hg emissions and deposition, as well as interactions dependent on land use and vegetation, forests, peat bogs etc. is highlighted in this study.

## 1 Introduction

Mercury (Hg) is a toxic element, present on Earth in different environmental compartments. Due to its chemical and physical properties Hg is a global pollutant, and in its monomethyl form hazardous to human and wildlife health. The main human methylmercury exposure pathway is through piscivorous fish consumption. Soluble inorganic $Hg^{II}$ compounds can be methy-

lated in the marine environment and enter the base of the food chain (Chen et al., 2016; Žagar et al., 2013; Oken et al., 2012; Storelli et al., 2010). Hg exists in atmosphere as Gaseous Elemental Mercury (GEM, $Hg^0$), Reactive Gaseous Mercury (RGM, $Hg^{II}_{(g)}$) and Particulate Bound Mercury (PBM, $Hg^{(p)}$). The term RGM describe all forms of Hg sampled using a KCl-coated denuder and analysed by CVAAFS (Landis et al., 2002; Gustin et al., 2015), the exact chemical nature of these compounds is still not known. There is also still some debate also over the oxidation and reduction mechanisms that occur in the atmosphere (Subir et al., 2011; Gustin et al., 2015). RGM is much less volatile and more water soluble than GEM, and thus is readily transferred to aquatic and terrestrial ecosystems by dry and wet deposition processes. Moreover, GEM concentrations in some regions are linked to large scale climatological phenomena (Carbone et al., 2016).

Since European Hg riverine discharges have been reduced greatly since the '70s, atmospheric deposition has become the most important source of Hg in the marine ecosystems.

In order to support the recent Minamata convention (http://www.mercuryconvention.org/), the GMOS (Global Mercury Observation System, http://www.gmos.eu/) project has set up a global Hg observation network, with the aim to work alongside existing networks for Europe, USA and Asia. The data obtained by GMOS has made it possible to fill in some of the gaps left by regional networks, by performing measurements in places which have until now not been studied, especially in the Southern hemisphere. In Europe the GMOS network has complemented the EMEP regional network with special measurements, such as the Med-Oceanor oceanographic campaigns in the Mediterranean Marine Boundary Layer (MBL, Gencarelli et al. (2014b); Sprovieri et al. (2010)) the ETMEP (European Tropospheric Mercury Experiment) series of Tropospheric measurements (Weigelt et al., 2016) and the fixed station at Longobucco, in the South of Italy (Sprovieri et al., 2016b). This station is currently the southernmost in Europe, and is in the centre of the Mediterranean Sea region. Compared to the north of Europe however southern and eastern Europe still have a lack of monitoring stations. Within GMOS global CTMs have been used to evaluate the intercontinental transport of anthropogenic Hg (De Simone et al., 2016), analysed different future Hg emission scenarios (Pacyna et al., 2016), source-receptor relationships (Travnikov et al., 2010) and Hg cycle in the Polar regions (Angot et al., 2016). Regional scale models have been used for a closer study of the processes that occur in specific areas at much higher spatial resolution than the global CTMs (Gencarelli et al., 2015; Bieser et al., 2014). In a recent review Ariya et al. (2015) gives a exhaustive summary of modelling progress and of the uncertainties still present concerning the atmospheric Hg cycle. To date only a limited number of model to model intercomparisons have been carried out (for the U.S. (Bullock et al., 2008, 2009; Zhang et al., 2012), for Europe (Ryaboshapko et al., 2007a, b), and for global models (Travnikov et al., 2010, 2016; AMAP/UNEP, 2013b)), where it was found that are often significant differences in Hg concentrations and deposition estimated by different models. Previous European studies (Ryaboshapko et al., 2007a, b) performed a model intercomparison for the year 1999, using 8 different models and data from 11 measurement stations with the aim to characterise the ability of CTMs to predict atmospheric Hg concentration and deposition fields.

The aim of this work is to analyse the influence of different processes affecting atmospheric Hg and quantify some of the uncertainties present in the regional Hg cycle (highlighted in the aforementioned model comparisons), in particular anthropogenic emissions speciation and the $Hg^0_{(g)}$ atmospheric oxidation mechanism. The atmospheric Hg cycle has been simulated over Europe for the year 2013, performing 14 different model sensitivity tests using the WRF/Chem-Hg model for the most part.

A number of further investigations were also performed using the CMAQ-Hg model, to gain an insight into model-to-model variation. The model sensitivity tests were conducted using different anthropogenic emission datasets, Br concentration input fields, Hg oxidation schemes and global CTMs to provide boundary condition input. The experimental results were compared with observations from 28 monitoring sites in Europe (23 from the EMEP network, 4 from the EMEP/GMOS network including the Longobucco station of the GMOS network), in order to validate model performance and investigate the influences of anthropogenic emission speciation and the $Hg^0_{(g)}$ atmospheric oxidation mechanism. A more detailed screening for some selected stations was performed, in order to investigate some anomalies in the sensitivity model results.

The work presented here was performed in the framework of the GMOS Mercury Modelling Task Force (MMTF, Travnikov et al. (2016)).

## 2 Methods

### 2.1 Models description and setup

The models used in this analysis both use a Lambert Conformal model domain covering Europe and the Mediterranean Sea, including part of the western North Atlantic Ocean, North Africa and the Middle East (see Fig. 1) with a horizontal resolution of $24 \times 24$ km, and 30 vertical levels. The online WRF/Chem-Hg model (Gencarelli et al., 2014a)) is a modified version of the WRF/Chem model (version 3.4 Grell et al. (2005)) which includes emissions, transport, atmospheric chemistry and deposition of Hg.

The Hg emissions in the model include online GEM evasion from the sea surface (based on the parametrisation of Wanninkhof (1992) and the methodology of Gårdfeldt et al. (2003), with Dissolved Gaseous Mercury concentrations of $150 \, \text{fmol} \, 1^{-1}$) and anthropogenic emissions from the AMAP (AMAP/UNEP, 2013a) and EDGAR (Muntean et al., 2014) inventories (see section 2.2). The gas phase chemistry of Hg and a parametrised representation of atmospheric Hg aqueous chemistry have been added to the RADM2 chemical mechanism using KPP (Sandu and Sander, 2006) and the WKC coupler (Salzmann and Lawrence, 2006), while the atmospheric physics and transport are solved by the WRF model core using the parametrisations described in Gencarelli et al. (2014a). Hg dry deposition is treated according to the approach developed by Wesely (1989) and calculated as described in Lin et al. (2006). Wet deposition (in-cloud and below-cloud scavenging) of Hg species has been implemented by adapting an already available module in WRF/Chem, based on the approach described by Neu and Prather (2012). Chemical Initial and Boundary Conditions (IC/BC) were taken from the ECHMERIT model Jung et al. (2009); De Simone et al. (2014, 2015, 2016) for Hg species, while boundary conditions for other chemical species were taken from MOZART-4 (Emmons et al., 2010).

The second model used is CMAQ-Hg (version 5.0.1), based on CMAQ (Byun et al., 1999) and modified by Bullock and Brehme (2002) and Gbor et al. (2006) to include chemistry, transport and deposition of GEM, GOM and PBM. This model was compiled with the multi-pollutant version of the CBM5 photochemical mechanism (Sarwar et al., 2008) (which includes Hg gaseous reactions with $O_3$, OH, $H_2O_2$ and $Cl_2$ as described by Lin and Tao (2003)) with th eEuler Backward Iterative solver and the AERO4 aerosol mechanism (Binkowski and Roselle, 2003). The CMAQ-Hg model uses offline meteorological fields

provided by the COSMO-CLM model (Rockel et al., 2008), processed by the Meteorology-Chemistry Interface Processor (MCIP v3.6). The same MCIP to calculate the dry deposition velocities of GEM and GOM. During the offline simulations cloud processes, including cloud attenuation of photolysis rates, convective and non-convective mixing and scavenging by clouds, aqueous-phase chemistry, and wet deposition were calculated as described in Liu and Zhang (2013). The chemical IC/BC were taken from the GLEMOS model (Travnikov et al., 2009). For further details on the models see Gencarelli et al. (2014a, 2015) for WRF/Chem-Hg and Bieser et al. (2014); Zhu et al. (2015) for CMAQ-Hg.

The main difference between the two models is in the feedback between chemical and meteorological dynamics: while in CMAQ the meteorological fields are provided as input (offline model), in WRF they are solved simultaneously with the chemistry, in the same time step (online model). Other major differences concern the parametrisations of some of the processes, for instance, GEM dry deposition, convective precipitation and GEM evasion from the sea surface (see Gencarelli et al. (2015) and Bieser et al. (2014) for details). Other differences result from the use of different BC sets and meteorological input.

Oxidation of Hg by bromine was implemented in some of the WRF experiments, using the off-line Br fields obtained from the p-TOMCAT (Yang et al., 2005, 2010) and GEOSCHEM (Parrella et al., 2012) models.

## 2.2 Modelled emissions

In order to analyse the effects of anthropogenic emissions speciation, amount and vertical distribution, the input from the two recent global anthropogenic Hg emission inventories were interpolated over the model grids and used in the sensitivity simulations.

The AMAP/UNEP 2010 (hereafter AMAP) inventory is available at a spatial resolution of 0.5° by 0.5° (AMAP/UNEP, 2013b), while the EDGARv4.tox1 2008 (hereafter EDGAR) has a spatial resolution of 0.1° by 0.1° (Muntean et al., 2014). Over the modelling domain the inventories differ in both emission totals and speciation ratio GEM:RGM:PBM as:

- 136.2 Mg y$^{-1}$ with GEM:RGM:PBM ratio 65:28:7 for AMAP, and

- 123.8 Mg y$^{-1}$ with 60:32:8 for EDGAR.

They also have different emission height distributions: AMAP uses three height classes (0-50, 50-150 and above 150 m) whereas EDGAR ranges into six classes (distributed between 0 and 800 metres, listed according with SNAP (Selected Nomenclature for Air Pollution) categories as used in De Simone et al. (2016)). The differences in the geographical distributions are shown in Fig. 2.

## 2.3 Simulations performed

Simulations were performed varying the emissions speciation, the atmospheric Hg oxidation mechanism, the bromine concentration field and the atmospheric process parametrisation. A total of 14 (9 with WRF and 5 with CMAQ) 12-month model simulations were conducted, as reported in Table 1, where experiments conducted using CMAQ are indicated by a $C$ subscript. The specific scopes of every particular experiment as:

- BASE – base case test, used as reference experiment. It refers to the model in the standard configuration, with AMAP anthropogenic emissions and Hg oxidation driven only by $O_3$ and OH for WRF/Chem-Hg and by $O_3$, OH, $H_2O_2$ and $Cl_2$ for CMAQ-Hg, as described in section 2.1.

- BASE2 – similar to BASE experiment, with the only change of Hg anthropogenic emission used. In fact in this case EDGAR Hg emissions are used.

- NOANT – hypothetical scenario, where all anthropogenic emissions are turned off, in order to highlight the influence of long-range transport on European areas. The same chemical mechanism of BASE experiment is used.

- NOCHEM - hypothetical scenario, where the chemical reactions of Hg are turned off. In this way there are not conversion of GEM in RGM, that imply a different distribution of Hg deposition respect the BASE experiment.

- ANTSPEC – hypothetical experiment where all Hg emissions are treated as GEM. With this experiment RGM and PBM emission are turned off, and considering the different chemical and physical properties respect the GEM the deposition can occur in far place respect the emission points. It would represent a lower bound on deposition from local anthropogenic sources and an upper bound on long-range transport of anthropogenic emissions because GEM has a much longer lifetime against deposition than RGM and PBM.

- O3CHEM/OHCHEM/BRCHEM1/BRCHEM2 - hypothetical experiments where only a singular reaction of Hg atmospheric was activated (only $O_3$, OH and Br respectively). The HgBr+OH rate constant is taken from the assumptions made in Holmes et al. (2010) (Global atmospheric model for mercury including oxidation by bromine atoms, Atmos. Chem. Phys., 10, 12037–12057). These sensitivity tests are not a direct comparison between the chemical mechanisms, but are an analysis of how much the atmospheric system change considering an singular Hg oxidant. The difference between BRCHEM1 and BRCHEM2 regard only the answer of the system at the change of amount of Br in the atmosphere.

A summary of the simulations performed is shown in Table 1. Some of these tests have been studied for other regions (e.g. Travnikov et al. (2016) and Bieser et al. (2016)) while many other studies have investigated Hg oxidation by Ozone or Br (Hynes et al., 2009; Subir et al., 2011, 2012; Weiss-Penzias et al., 2014).

## 2.4 Measurement networks

Model results have been compared with observations of Total Gaseous Mercury (TGM, $Hg_{(g)}^0 + Hg_{(g)}^{II}$) and Hg wet deposition from 28 EMEP and GMOS measurement sites as shown in Table 2. Of these, 13 measured both TGM air concentrations and Hg in precipitation, 4 measured only TGM and 11 only Hg in precipitation (Fig. 1). Comparison was made between monthly averaged values of TGM observations and monthly Hg in precipitation (Aas and Bohlin-Nizzetto, 2015; D'Amore et al., 2015). Monthly averages were used because the measurement frequency was not the same at all the sites.

## 3 Results

The principal differences between the models used concern the parametrisations of some of the processes, for instance, GEM dry deposition, convective precipitation and GEM evasion from the sea surface (see Gencarelli et al. (2015) and Bieser et al. (2014) for details). Other differences result from the use of different BC sets and meteorological input. Despite these differ-
ences, the base cases (BASE and BASE$_C$) give similar Hg deposition totals of 219 Mg y$^{-1}$ in the WRF BASE and 208 Mg y$^{-1}$ in the CMAQ BASE$_C$ experiments (table 3). The differences in deposition parametrisations does have an effect on the ratio of dry to wet Hg deposition however. While dry and wet deposition are almost equal in the WRF simulations (wet 49%, dry 51%), the dry deposition in CMAQ is more than twice the wet (69% dry and 31% wet), see table 3 and figures 3 and 4 for details. There are major differences in the spatial distribution of the Hg deposition, wet deposition in WRF is greater over continental
Europe, the North Sea and around Iceland, while in CMAQ deposition is highest over the Alps and along the Balkan coast. Although both models simulate higher dry deposition over land than the sea the distribution in CMAQ is more even than that simulated by WRF, which has quite distinct regions and hot spots of high deposition (Fig. 3).

### 3.1 Modelled and Observed Hg species comparison

The skill of the WRF/Chem-Hg and CMAQ-Hg model in reproducing the modelled Hg concentrations, deposition fluxes and precipitation fields has been described in previous studies (see Gencarelli et al. (2015), Bieser et al. (2014) and references therein). Thus, the principal aim of this study is to analyse the performance of models in terms of validation of the sensitivity tests, also comparing the results of all the simulations performed with the available observations reported in Sect. 2.4.

Generally for GEM atmospheric concentrations there is a general underestimation in the WRF model simulations and an
overestimation in CMAQ model simulations. For wet deposition values the CMAQ model tends overestimate the observations, especially in Scandinavia, England and at Longobucco. On the other hand the WRF model has different characteristics: in Scandinavia the observations are always overestimated when compared to the rest of the domain, in the BASE2 experiment the greatest overestimation occurs while in the ANTSPEC experiment there is a general underestimation almost everywhere (given the lack of RGM emissions it is not surprising that the deposition is lower in this experiment).

Overall the agreement of the comparison between base model (BASE, ANTSPEC, BASE$_C$, ANTSPEC$_C$ and BASE2) results and observations obtained (both for TGM concentrations and Hg in wet deposition) at all stations are broadly in agreement with previous studies (e.g., Ryaboshapko et al. (2007b)). Comparing modelled and observed values of TGM concentrations the ratio of annual pairs Model-Observation is within 30% in almost all stations for the BASE, ANTSPEC, BASE$_C$, ANTSPEC$_C$, and BASE2 experiments, while an obvious underestimation occurs in NOANT experiment(Fig. 5a). It is however interesting to look
at cases where the model to observation ratio is different in order to highlight the differences which are found in the sensitivity tests and in different locations. In the central, DE03 (Schauinsland) and southern, DE08 (Schmücke), German stations, the BASE and BASE2 experiments underestimate the observed annual average TGM concentration by 1.75 and 1.65 ng m$^{-3}$ respectively, while the ANTSPEC experiment shows better agreement. Contrarily the relatively nearby station at Kosetice

(CZ03, a rural location in the Czech Republic) the TGM concentrations are overestimated in the ANTSPEC experiment, while the BASE and BASE2 simulations show good agreement. In this station an annual average of $1.24\,\text{ng m}^{-3}$ was observed.

The DE03, DE08 and CZ03 sites are the most central European continental sites with available observations, and are characterised by an elevated contribution from local Hg emissions with respect to the contribution from long-range transport (Gencarelli et al., 2015). In Gencarelli et al. (2015) local sources are those within the domain, including anthropogenic emissions and evasion from the sea surface, while long-range sources are those from the boundary conditions obtained from the global model. The strongest influence of local emissions was found at the CZ03 station, as suggested by the large overestimation of GEM concentrations in the ANTSPEC experiment ($\simeq 37\,\%$, which instead was not the case in the BASE experiment).

This is due to the emissions from Chlor-alkali plants, which are still important sources in some parts of central Europe (Wang et al., 2012), while the different behavior seen at DE03 and DE08 is most likely due to a local emission process or processes, possibly of non-anthropogenic origin as argued in Siudek et al. (2016) for a study of forested Polish sites. The German sites used in this study are in mountainous and hilly forested regions (DE03 in the Black Forest - 1205 m asl - and DE08 in the Thuringian Forest, 937 m asl) are the two sites where the model underestimation is greatest.

At the GB48 station (Auchencorth Moss) and the coastal site of Niembo, ES08, the models fail to reproduce the low annual average TGM concentrations of 0.89 and $0.46\,\text{ng m}^{-3}$ respectively. At the GB48 site the disagreement between the models and observations can be attributed to local effects, as suggested by Drewer et al. (2010) in their study of greenhouse gas fluxes at the site. In fact this site is located in a peat bog, and the observed TGM values are very different from sites at similar latitudes such as DK01, Færøerne and IE31, Mace Head where the annual average TGM concentrations were 1.56 and $1.49\,\text{ng m}^{-3}$ respectively, and where the models are able to reproduce the observations.

Overall the overestimation of GEM concentrations using WRF/Chem-Hg is greater in the OHCHEM experiment due to a lower rate of $\text{Hg}^{0}_{(g)}$ oxidation and lower in NOANT because there are no anthropogenic emissions.

The monthly comparison between the observed and modelled concentrations are shown in Fig. 6a and Fig. 7a (only the principal experiments are shown). There is a clear overestimation of monthly concentrations by CMAQ, particularly during summer. Only small differences occur changing the anthropogenic emissions inventory (BASE2), while the differences when the speciation (ANTSPEC) and the chemical mechanism (BRCHEM1) are changed and are much more evident. Decreasing the uncertainty in flue gas speciation would be a great advantage in modelling the atmospheric Hg cycle.

Comparing modelled and observed values of wet deposition fluxes gives a ratio of annual pairs Model-Observation within a factor of 2 in most stations (15 of the 24 stations), while in 23 of the 24 stations it is within a factor of 5 (see Fig. 5b). The outlier is the Valentia Observatory (IE01) in South-West Ireland: the annually averaged observed Hg deposition is $1.70\mu\text{g m}^{-2}\,\text{month}^{-1}$, which is high with respect to the median of $0.31\mu\text{g m}^{-2}\,\text{month}^{-1}$ and the average of $0.46\mu\text{g m}^{-2}\,\text{month}^{-1}$ in all stations, ($1.70\mu\text{g m}^{-2}\,\text{month}^{-1}$ is approximately the 97[th] percentile). Moreover, the underestimation is more notable in WRF (ratio $\simeq 0.10$) than CMAQ ($\simeq 0.40$).

Overall in the BASE and ANTSPEC experiment slight overestimates were found, while the results from CMAQ experiments show higher Hg wet deposition fluxes than in the WRF/Chem-Hg experiments. In Sprovieri et al. (2016a) high values of wet deposition in Råö (SE14) and Pallas (FI36) stations were found, compared to the other European stations in the GMOS net-

work. The model results reflect this result, with high deposition fluxes in the North of Europe. In these stations, as in all of Scandinavia, the oxidation mechanism makes a great difference, see the BRCHEM1 and BRCHEM2 experiments.

## 3.2 Emissions speciation

Recently, in order to study the impact of Hg anthropogenic emission speciation on Hg deposition and its global cycle, some modelling studies have made use of modified anthropogenic emission inventories, either in terms of emission totals or in terms of the emission speciation (Selin et al., 2008; Amos et al., 2012; Horowitz et al., 2014)). For example Bieser et al. (2014) (for Germany) and Kos et al. (2013) (for the U.S.) obtain the best agreements between observations and model results when respectively assuming no RGM emissions, and using a modified emission speciation ratio, 90:8:2 instead of 50:40:10

(GEM:RGM:PBM, see Sect. 2.2). Gencarelli et al. (2015) compared the Hg deposition over Europe using the two most recent AMAP/UNEP inventories, showing that the lower emissions in 2010 resulted in lower simulated deposition fluxes, but that the deposition reduction was proportionally less than the emission reduction within the domain. With the experiments performed it was possible to estimate the impact of Hg anthropogenic emission speciation on Hg deposition. Specifically the results of the BASE, BASE2, ANTSPEC, NOANT, BASE$_C$ and ANTSPEC$_C$ simulations in Table 1 were compared (Fig. 8 and Fig. 10). The

BASE simulation used the AMAP Hg emissions (136.2 Mg y$^{-1}$, GEM:RGM:PBM 65:28:7) while the BASE2 simulation used the EDGAR emissions (123.8 Mg y$^{-1}$, ratio 60:32:8). The difference in emitted Hg over the modelling domain makes little difference in terms of the TGM concentrations and the wet deposition fluxes at the monitoring stations, see Fig. 5. However, over the whole domain the total Hg deposition is $\simeq 20\%$ less using the EDGAR inventory, as shown in Table 3. The deposition pattern changes, often in areas characterised by elevated Hg emissions were decreased deposition in BASE2 was found with

respect to the BASE experiment (where the ratio BASE/BASE2 is <1 in Fig. 8). This difference is very marked over the North and Baltic Seas, while it is almost negligible over the Mediterranean and Northern Atlantic. This effect is prevalently due to the difference in the vertical distribution of the emissions in the two experiments, because the total Hg emitted is very similar, there is only a 9% difference, contrarily to Gencarelli et al. (2015) where the same vertical distribution but different emissions totals resulted only in a small change in deposition. The change in emission vertical distribution prevalently affects dry deposition

processes, decreasing by 28% between the BASE - BASE2 simulations, against a 13% decrease in wet deposition (Fig. 3 and Fig. 4). In the BASE2 simulation deposition is noticeably lower over the Balkans, Carpathians and the lowlands of northern Germany, while Hg deposition is higher over the Skagerrak strait (which links the North and Baltic Seas).

  The ANTSPEC and ANTSPEC$_C$ simulations isolate the role of Hg emission speciation. In these simulations all emissions were considered to be Hg$_{(g)}^0$. Overall the simulations show an increase in the GEM concentration fields and a decrease in wet

deposition, leading to improved agreement with the GEM/TGM observations in the ANTSPEC simulation (however this is less evident in ANTSPEC$_C$). In ANTSPEC$_C$ improved agreement was obtained for wet deposition fluxes in some central (CZ03 and SI08) and northern (GB48, SE11 and SE14) monitoring sites. Total Hg deposition over the modelling domain decreased by 20% in WRF/Chem-Hg and by 22% in CMAQ. Dry deposition is particularly affected, see Table 3; with 28 and 26% decreases in dry deposition (WRF and CMAQ respectively) compared to a 13 and 14% decrease in wet deposition.

In the ANTSPEC experiment the deposition decreases in comparison to BASE, in particular the dry decreases more than the wet. RGM and PBM deposit more rapidly than GEM and so deposit in proximity to their emission sources where the air concentrations are higher. Clearly dry deposition can occur at any time while wet deposition requires precipitation. With all Hg emissions releases treated as GEM in ANTSPEC the dry deposition decreases most as a result of the lack of direct emissions of RGM and PBM. The areas most affected by changing the emission speciation are obviously near major sources, but also over the Aegean and the Black Seas, the English Channel and the North German coast. Considerable influences were found also over the Mediterranean, the North and Baltic Sea and the rest of Europe, while very little difference is seen over the Norwegian Sea and around Iceland, only minor differences were registered in DK01 station.

However the contribution of anthropogenic emissions is crucial. In fact, the complete exclusion of anthropogenic emissions (the NOANT experiment) cannot reproduce the TGM concentrations and wet deposition fluxes, they are clearly underestimated, and total Hg deposition is only one third of that when anthropogenic emissions are included. The NOANT experiment is a hypothetical scenario, but it allows the contribution of anthropogenic emissions to total deposition, which is roughly 2/3, not counting the fact that natural emissions from the oceans are in part previously deposited Hg from anthropogenic sources. A number of policy scenarios were used during the GMOS project to estimate future trends in the anthropogenic emission of Hg (Pacyna et al., 2016). Pacyna et al. (2016) describes the results of modelling studies using these scenarios to assess Hg concentration and deposition fields, for present (2013) and future anthropogenic (2035) Hg emissions.

### 3.3 Mercury oxidation

In order to highlight the differences due to the gas phase Hg oxidation mechanism employed various simulations were performed isolating a single oxidant in the model chemical mechanism.

The importance of the chemical reactions has been emphasised by considering the variations between BASE and NOCHEM experiments (and $BASE_C$ and $NOCHEM_C$), where no chemical interactions in the atmosphere were considered. In this experiment the RGM and PBM fields in the model domain are due prevalently to anthropogenic emissions, the influence BC on RGM and PBM is relatively minor over the model domain. Only very small changes in TGM air concentrations were found (there is slight increase $\simeq 1$ % in WRF, $\simeq 3$ % in CMAQ), while RGM and Hg deposition decrease appreciably (by 83% RGM, 25% wet and 73% dry in WRF, and 42%, 32% and 46% in CMAQ).

This result shows the net reduction in deposition when setting the Hg antrhopogenic emissions to zero, and provides a limit to the deposition due to natural emissions. With the exception of stations CZ03 and LV01 deposition is underestimated everywhere (especially in northern Europe, and at FI36 and GB13 above all). Agreement within a factor of 2 was found only in some stations in Central Europe (e.g., CZ03, DE02, DE09, LV01), demonstrating the importance of anthropogenic emissions speciation in these particular areas, with respect to BC and atmospheric oxidation.

A number of studies have shown the importance of $O_3$, and the OH radical, and also reactive halogen compounds in the atmospheric oxidation of Hg to form more readily deposited $Hg^{II}$ compounds (see Ariya et al. (2015) and references therein). Despite the theoretical doubts of significance of GEM oxidation under atmospheric conditions by $O_3$ and OH radical, that atomic Br is of great relevance to the atmospheric oxidation of GEM is certain (Hynes et al., 2009; Subir et al., 2011, 2012;

Weiss-Penzias et al., 2014). In particular, for the HgBr[*] intermediate, Dibble et al. (2012) has shown the potential importance of reactions with $NO_2$, HOO, ClO, and BrO.

While Subir et al. (2011) summarises the experimental and theoretical uncertainties in the calculation of the rate constants of these reactions (and also discusses the implications for CTMs), this study demonstrates the effect of individual oxidants on

tropospheric Hg concentrations and deposition, isolating the individual contributions and comparing them with the monthly wet deposition observed at the measuring stations. Based on the ANTSPEC experiment the three main GEM oxidants have been studied individually in the experiments O3CHEM, OHCHEM and BRCHEM1 (and BRCHEM2). Compared to the BASE case the emission scenario is different, all anthropogenic emissions were considered to be GEM, thus the RGM involved in the deposition process is solely the result of atmospheric oxidation processes.

In this way it is possible to estimate the extent of individual reactions on Hg oxidation and its deposition. The simulations are unrealistic because these reactions most likely occur simultaneously in the atmosphere (as in BASE and ANTSPEC cases), but these experiments are a potentially useful way to provide information on the principal oxidant pathways. The O3CHEM and OHCHEM experiment (executed BC from ECHMERIT) yield quite different results, both experiments are very different than observations. The OHCHEM experiment leads to the production of only small amounts of RGM $\simeq$40% less than ANTSPEC

over the whole domain and reduced deposition $\simeq$ 19 % less than ANTSPEC. Consequently GEM concentrations are higher $\simeq$ 35 %. As described in Subir et al. (2011) the mechanism of this reaction is unclear, and there are a number of different rate constants reported. In this study the Sommar et al. (2001a) rate constant was used, however, alone this oxidation pathway cannot explain the observed deposition, wet deposition fluxes are underestimated at all measuring stations. The underestimation is lowest in the southernmost stations, PT06 and LONG. In the O3CHEM experiment only GEM oxidation by $O_3$ is considered

(Hall, 1995). The results of this experiment show very high GEM concentrations (the ratio of GEM concentrations in O3CHEM and in BASE is $\simeq$ 1.33) but low RGM concentrations (ratio $\simeq$ 0.25) at ground level. Also the total depositions is underestimated (ratio $\simeq$ 0.24). As above, this reaction alone is not sufficient to represent oxidation and the deposition of Hg over Europe. Individually, oxidation by $O_3$ and OH do not give concentrations and fluxes comparable with the BASE case.

Using fixed BC (as in OHCHEM$_C$ and O3CHEM$_C$) the two simulations give very similar deposition (Fig. 8, Table 3). A

decrease compared to the BASE$_C$ case was found, but the effect of BC is dominant and the differences in the oxidation mechanisms are not appreciable.

The BRCHEM experiments provide more interesting results, more RGM is formed at ground level, the BRCHEM1/BASE ratio is $\simeq$ 1.63, BRCHEM2/BASE $\simeq$ 1.70, but the overall Hg wet deposition is lower than the base simulation (ratio $\simeq$ 0.44 and $\simeq$ 0.43 respectively). The comparison of model TGM to observations is within a factor of 2 in 16 of the 24 stations, which is the

best result for the set of the oxidation mechanism experiments (O3CHEM, OHCHEM, BRCHEM). A slight overestimation was found in the stations bordering the Baltic Sea and Mediterranean Sea. In fact the atmospheric Br concentrations in the upper troposphere and the MBL are the subject of much scientific interest (e.g., Subir et al. (2011); Hedgecock and Pirrone (2004); Saiz-Lopez and von Glasow (2012); Glasow et al. (2004)). In order to analyse the effect of different Br input (total amount and spatial distribution), the Br concentration fields from the off-line three-dimensional models p-TOMCAT (Yang et al., 2005)

and GEOSCHEM (Parrella et al., 2012) were interpolated on to the WRF domain. For every month a daily mean profile of

Br was obtained and used for BRCHEM1 (p-TOMCAT) and BRCHEM2 (GEOSCHEM), where a two-step GEM oxidation process which proceeds firstly by reaction with Br to form unstable diatomic HgBr[*] was implemented (see Gencarelli et al. (2015) and references therein for details). Notable differences in the two Br input fields exist in the Planetary Boundary Layer (PBL) and indeed in the whole atmospheric column exist. In fact, in the PBL, the amount of Br in the BRCHEM2 experiment is $\simeq 4.5$ times greater than that in BRCHEM1, with differences ranging from a factor of 6 (in the cold months) to roughly 4 (in the summer months) over the modelling domain. These differences are also visible observing the vertical longitudinal profiles of annual mean concentrations of Br in Fig. 9, where Br in BRCHEM2 is greater than Br in BRCHEM1 in the first 3 km. The Br in BRCHEM1 is greater than BRCHEM1 in the range between 12 and 15 km, while at other elevations there are not such large differences.

Despite this large discrepancy in Br input the Hg deposition flux is largely unaffected. Total Hg deposition in BRCHEM1 is 3% greater than BRCHEM2 (2 % wet and 3.7 % dry), due to greater concentrations of Br in areas with high rainfall in BRCHEM1. The spatial distribution of the deposition is different in the two experiments, especially over North Africa, the English Channel and the Western Mediterranean. The RGM concentrations in the PBL are also slightly different (RGM in BRCHEM1 about is 4% lower than BRCHEM2, as shown in Fig. 8 and Table 3). Model to observation comparison for TGM does not change between BRCHEM1 and BRCHEM2, and the model underestimates wet deposition in both cases. In BRCHEM1 the underestimation is evident (ratio BRCHEM1/Observations $\simeq 0.77$) while in BRCHEM2 the model results and observations are closer (ratio $\simeq 0.90$). Thus, increasing the Br concentration gives model results closer to observations, as shown also in Shah et al. (2015), who tripled the Br and BrO concentrations in GEOS-CHEM model (with respect to Parrella et al. (2012), where an underestimation of 30% was found for BrO concentrations compared to the global mean obtained by satellite observations) to model the NOMADSS field campaign (Gratz et al., 2015). Differently to the Shah et al. (2015) study, RGM reduction is not implemented here, for the purposes of this paper it was necessary to standardise the experiments as far as possible. Only an aqueous phase reduction was implemented in CMAQ following Si and Ariya (2008).

A comparison of the annual mean RGM concentration in the first model level simulated in the various sensitivity runs is shown in figure 10. While the BASE and BASE$_C$ show a similar pattern, the ANTSPEC simulation (figure 10 first row) gives lower concentrations with no areas of elevated RGM concentrations. The ratio of the RGM concentration in the BASE2 and BASE simulations is strongly dependent on the distribution of the anthropogenic emissions, while the mean RGM concentrations in the BASE simulation are always greater then in NOANT, especially in central Europe. The opposite occurs when comparing BASE and NOCHEM, where the ratio is ~1 central Europe and <1 over the Mediterranean and the northern part of the domain. In the O3CHEM simulation the RGM concentrations are greater than those in ANTSPEC over the whole modelling domain, while in the OHCHEM and BRCHEM the ratio changes over the domain: in OHCHEM RGM is relatively higher over land and the Mediterranean Sea, while in BRCHEM1 RGM is lower over Mediterranean region and the Black Sea, but higher over the eastern and northern parts of the domain. In the BRCHEM2 the relative decrease in RGM is found to be mostly over the eastern part of the domain and Scandinavia. Using fixed BC (the third column of figure 10) the simulations show decreased RGM concentrations with respect BASE$_C$, especially in the areas characterised by significant anthropogenic emissions.

## 4   Conclusions

This work was performed to analyse the influence of several processes which determine the atmospheric Hg cycle and quantify some of the uncertainties present over a European modelling domain. The output of 14 model sensitivity tests were compared between themselves and with available measurements from 28 monitoring sites. The base experiments (BASE and $BASE_C$)

furnish model results roughly in accord with measurements of TGM concentrations and wet deposition fluxes, and agree with the results of observations reported in (Sprovieri et al., 2016a), with higher Hg deposition fluxes in the North of Europe.

In the model results the quantity, speciation and vertical profile of anthropogenic Hg emissions is crucial: over the whole model domain the vertical distribution of Hg emissions has a large influence on the Hg deposition fields. In addition to the areas near the principal anthropogenic emission sources, the areas of Aegean and the Black Seas, the English Channel, the Skagerrak

strait and the North Germany coast are largely influenced by the characteristics of European Hg emissions speciation more than they are by the total amount.

Using a reaction mechanism with GEM oxidation by only $O_3$ or OH greatly underestimate the observed deposition in precipitation. Whereas using a mechanism with Br as the GEM oxidant produces more RGM at ground level, but the overall Hg wet deposition are lower than the BASE simulation, which employs both $O_3$ and OH in the oxidation mechanism. Nonetheless good

agreement between the model and observations was found, especially in the stations bordering the Baltic and Mediterranean Seas. The Hg deposition was only slightly affected by the choice of Br input fields, quadrupling the Br air concentrations in the PBL resulted in a change of only of 3% in total deposition in accord with the results of Shah et al. (2015) for the U.S. The necessity to perform continuous measurements of speciated Hg in order to refine model oxidation mechanisms is clear. Moreover, the necessity to investigate more thoroughly local influences on Hg emissions and deposition, as well as interactions

dependent on land use and vegetation, forests, peat bogs etc. should be investigated in future studies.

# 5 Tables and figures

**Table 1.** Simulations performed

| exp | CTM | BC | Ant. Emiss | Description |
|---|---|---|---|---|
| BASE | WRF/Chem-Hg | ECHMERIT | AMAP | Base experiment, performed as in Gencarelli et al. (2015) |
| ANTSPEC | WRF/Chem-Hg | ECHMERIT | AMAP | RGM and PBM emissions as GEM |
| NOANT | WRF/Chem-Hg | ECHMERIT | AMAP | No Hg anthropogenic emissions |
| NOCHEM | WRF/Chem-Hg | ECHMERIT | AMAP | No Hg chemical reactions |
| O3CHEM | WRF/Chem-Hg | ECHMERIT | AMAP | As ANTSPEC, but with only $O_3$ GEM oxidation (Hall, 1995) |
| OHCHEM | WRF/Chem-Hg | ECHMERIT | AMAP | As ANTSPEC, but with only OH GEM oxidation (Sommar et al., 2001b) |
| BASE2 | WRF/Chem-Hg | ECHMERIT | EDGAR | As BASE, but with anthropogenic emissions from EDGAR inventory |
| BRCHEM1 | WRF/Chem-Hg | ECHMERIT | AMAP | As ANTSPEC, bu with only Br two step GEM oxidation, Br from p-TOMCAT |
| BRCHEM2 | WRF/Chem-Hg | ECHMERIT | AMAP | As BRCHEM1, but with Br input from GEOS-Chem model |
| $BASE_C$ | CMAQ-Hg | GLEMOS | AMAP | Base CMAQ experiment |
| $ANTSPEC_C$ | CMAQ-Hg | GLEMOS | AMAP | RGM and PBM emissions as GEM |
| $NOCHEM_C$ | CMAQ-Hg | GLEMOS | AMAP | No Hg chemical reactions |
| $O3CHEM_C$ | CMAQ-Hg | GLEMOS | AMAP | As $ANTSPEC_C$, but with only $O_3$ GEM oxidation |
| $OHCHEM_C$ | CMAQ-Hg | GLEMOS | AMAP | As $ANTSPEC_C$, but with only OH GEM oxidation |

*Acknowledgements.* We are grateful to the WRF/Chem developers and to the NCAR ESL Atmospheric Chemistry Division for making the WRF/Chem and the WRF/Chem preprocessor codes freely available. We gratefully acknowledge EMEP for maintaining and making available the database of monitoring station data. We thank Noelle Eckley Selin and Shaojie Song at the Massachusetts Institute of Technology, Cambridge,for completion of GEOSCHEM Br input, used in the BRCHEM2 experiment. The research was performed in the framework of the EU project GMOS (FP7 - 265113), the National Reference Centre for Mercury (CNRM, Italy) and the STM program of the Italian CNR.

**Table 2.** List of observation points

| CODE | Name | Network | TGM | WD |
|------|------|---------|-----|-----|
| BE14 | Koksijde | EMEP | | • |
| CZ03 | Kosetice | EMEP | • | • |
| DE02 | Waldhof | EMEP | • | • |
| DE03 | Schauinsland | EMEP | • | • |
| DE08 | Schmücke | EMEP | • | • |
| DE09 | Zingst | EMEP | • | • |
| DK01 | Færøerne | EMEP | • | |
| ES08 | Niembro | EMEP | • | • |
| FI36 | Pallas | EMEP/GMOS | • | • |
| GB13 | Yarner Wood | EMEP | | • |
| GB17 | Heigham Holmes | EMEP | | • |
| GB36 | Harwell | EMEP | | • |
| GB48 | Auchencorth Moss | EMEP | • | • |
| GB91 | Banchory | EMEP | | • |
| IE01 | Valentia Obs. | EMEP | | • |
| IE31 | Mace Head | EMEP/GMOS | • | |
| LV01 | Rucava | EMEP | | • |
| NL91 | De Zilk | EMEP | | • |
| NO01 | Birkenes | EMEP | • | • |
| NO90 | Andoya | EMEP | • | |
| PL05 | Diabla Gora | EMEP | • | |
| PT04 | Monte Velho | EMEP | | • |
| PT06 | Alfragide | EMEP | | • |
| SE05 | Bredkalen | EMEP | • | • |
| SI08 | Iskrba | EMEP/GMOS | | • |
| SE11 | Vavihill | EMEP | • | • |
| SE14 | Råö | EMEP/GMOS | • | • |
| LONG | Longobucco | GMOS | • | • |

| experiment | Wet | Dry | Total |
|---|---|---|---|
| BASE | 108.0 | 111.0 | 219.0 |
| ANTSPEC | 88.0 | 32.0 | 120.0 |
| NOANT | 52.1 | 19.2 | 71.3 |
| NOCHEM | 26.5 | 81.4 | 107.9 |
| O3CHEM | 38.5 | 13.9 | 52.4 |
| OHCHEM | 27.8 | 14.3 | 42.1 |
| BASE2 | 94.4 | 79.9 | 174.3 |
| BRCHEM1 | 63.2 | 33.2 | 96.4 |
| BRCHEM2 | 60.2 | 32.0 | 92.2 |
| $BASE_C$ | 65.5 | 142.7 | 208.2 |
| $ANTSPEC_C$ | 56.5 | 105.7 | 162.2 |
| $O3CHEM_C$ | 52.2 | 99.1 | 151.3 |
| $OHCHEM_C$ | 50.5 | 98.9 | 98.9 |
| $NOCHEM_C$ | 21.2 | 65.2 | 86.4 |

**Table 3.** Wet and dry annual deposition (Mg).

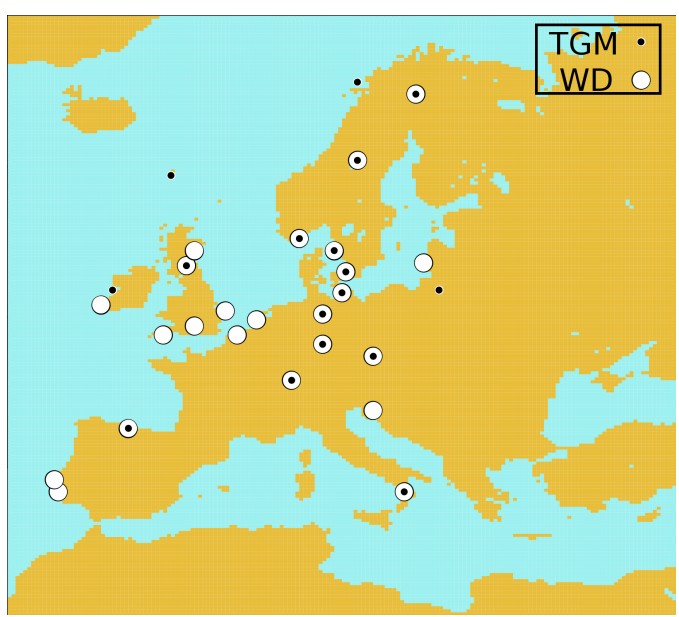

**Figure 1.** The model domain and location to the measurement stations (white points for wet deposition (WD) and black points for TGM air concentrations).

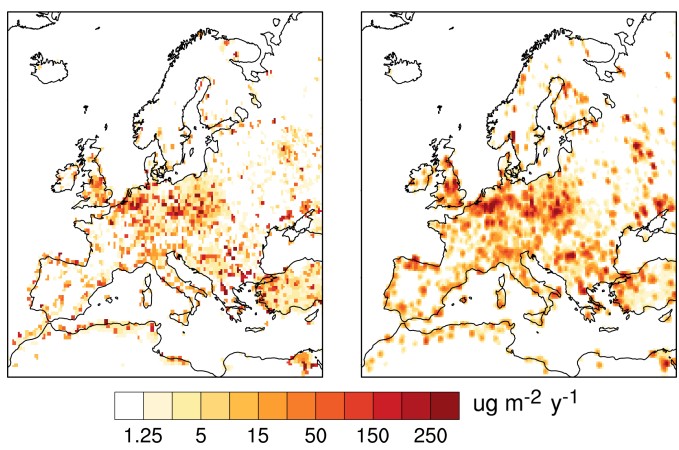

**Figure 2.** The anthropogenic Hg emissions used in the model experiments: AMAP (left panel) and EDGAR (right panel)

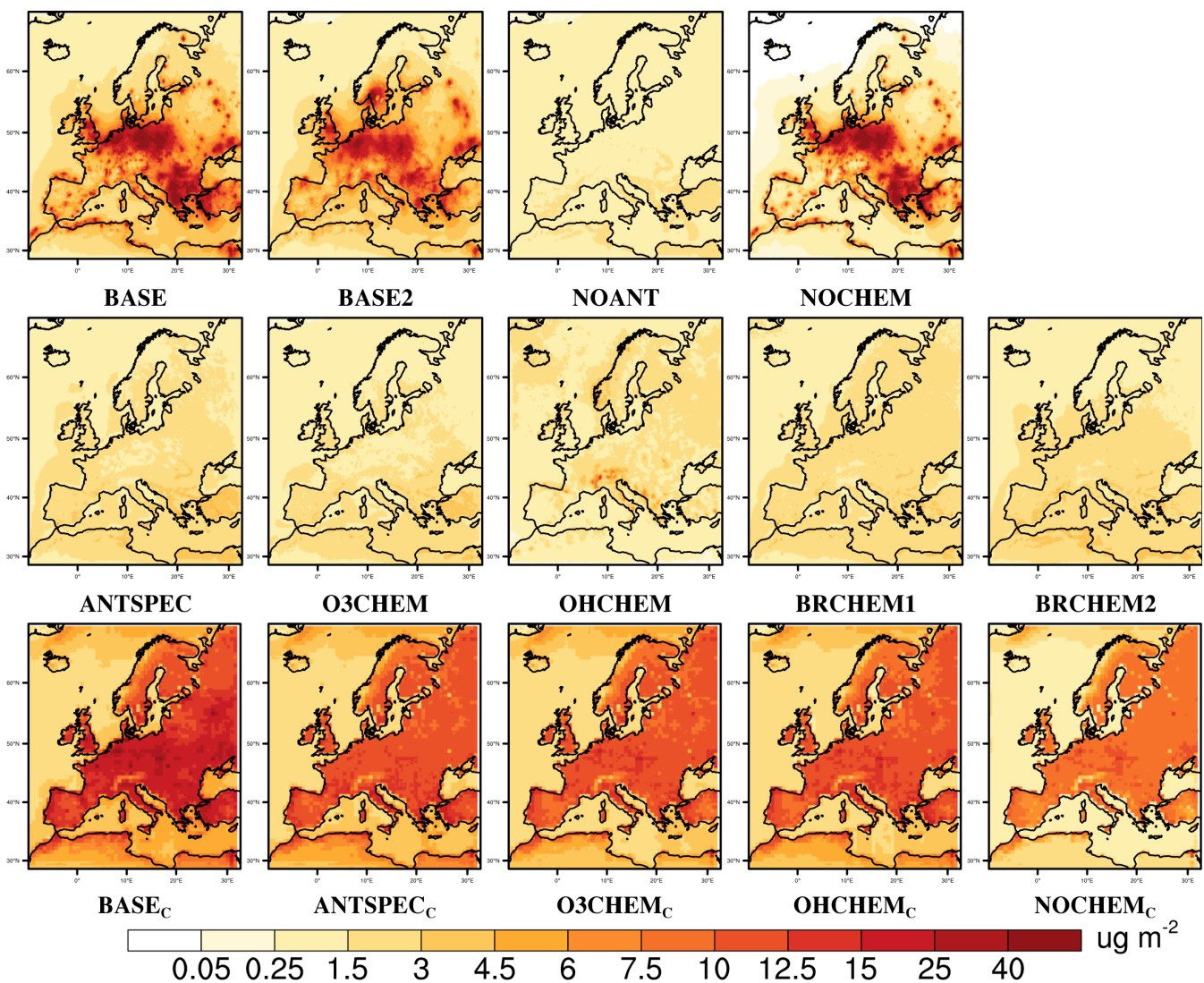

**Figure 3.** The total Hg dry deposition in the model experiments performed.

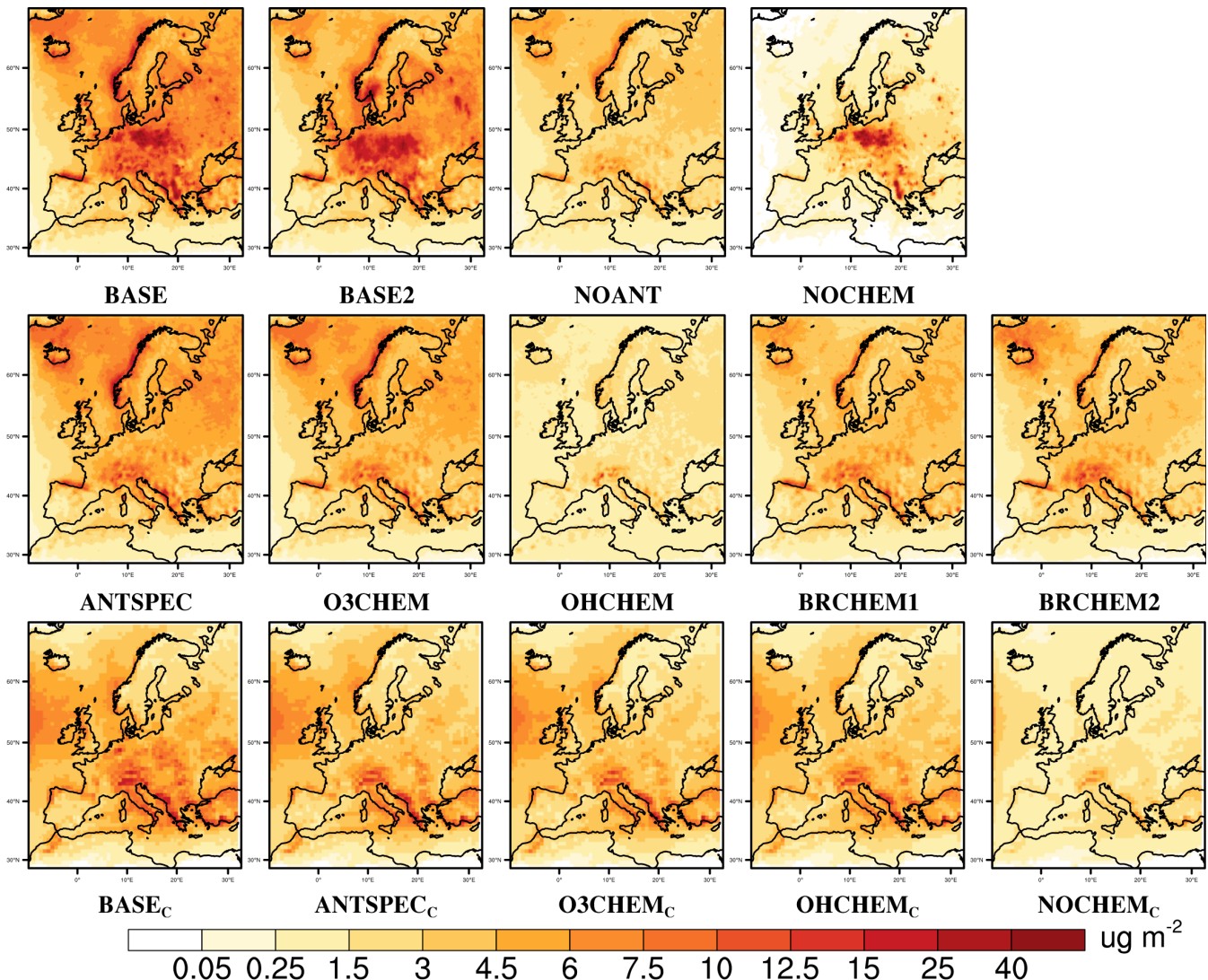

**Figure 4.** The total Hg wet deposition in the model experiments performed.

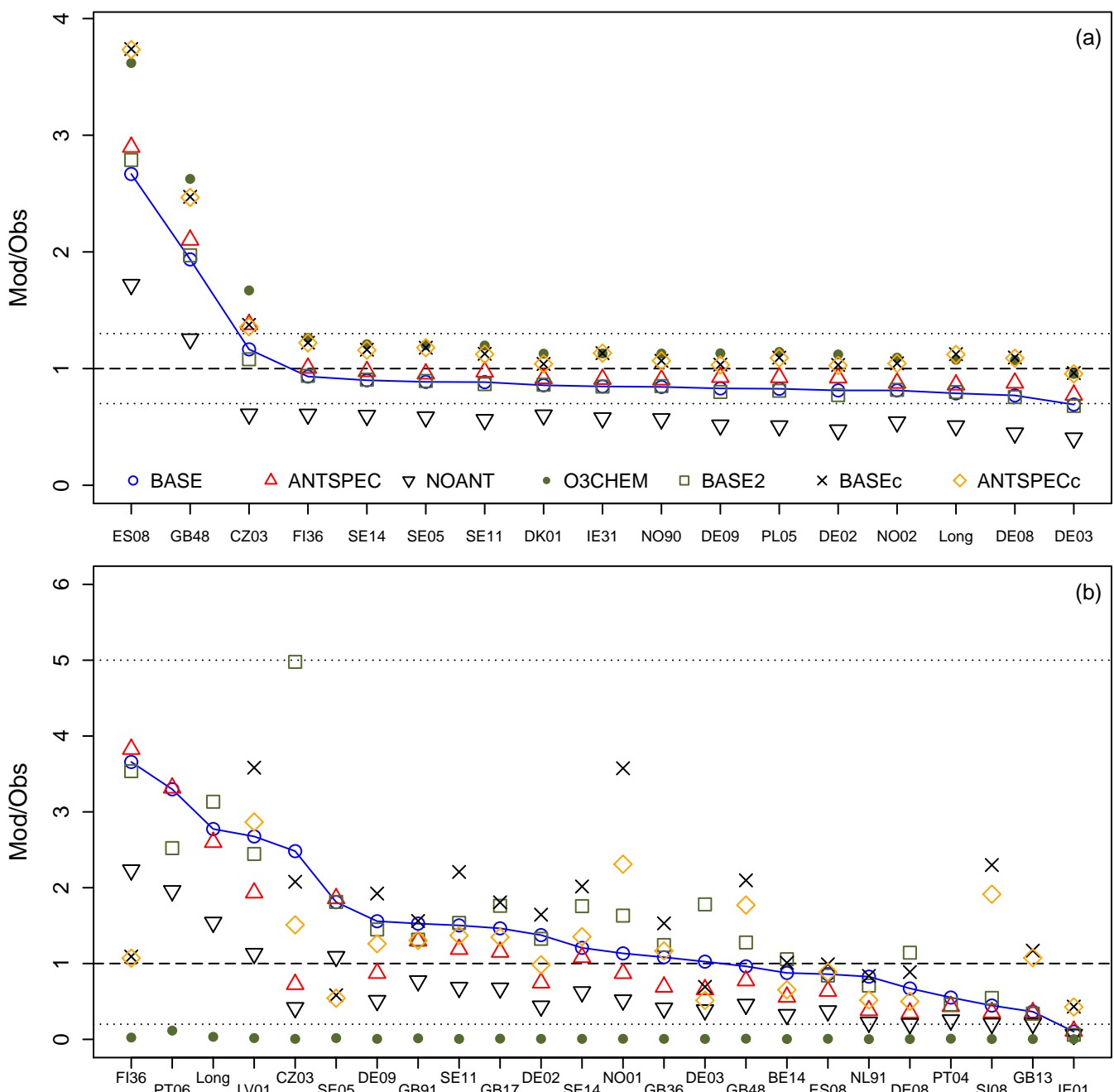

**Figure 5.** Ratio of the modelled and observed values at the measurement station points. Upper panel (a) for GEM concentrations and lower panel (b) for wet deposition fluxes. Horizontal lines represent perfect agreement (dashed line, ratio = 1) and good agreement intervals (dotted lines, ± 30 % for GEM, factor 5 for WD).

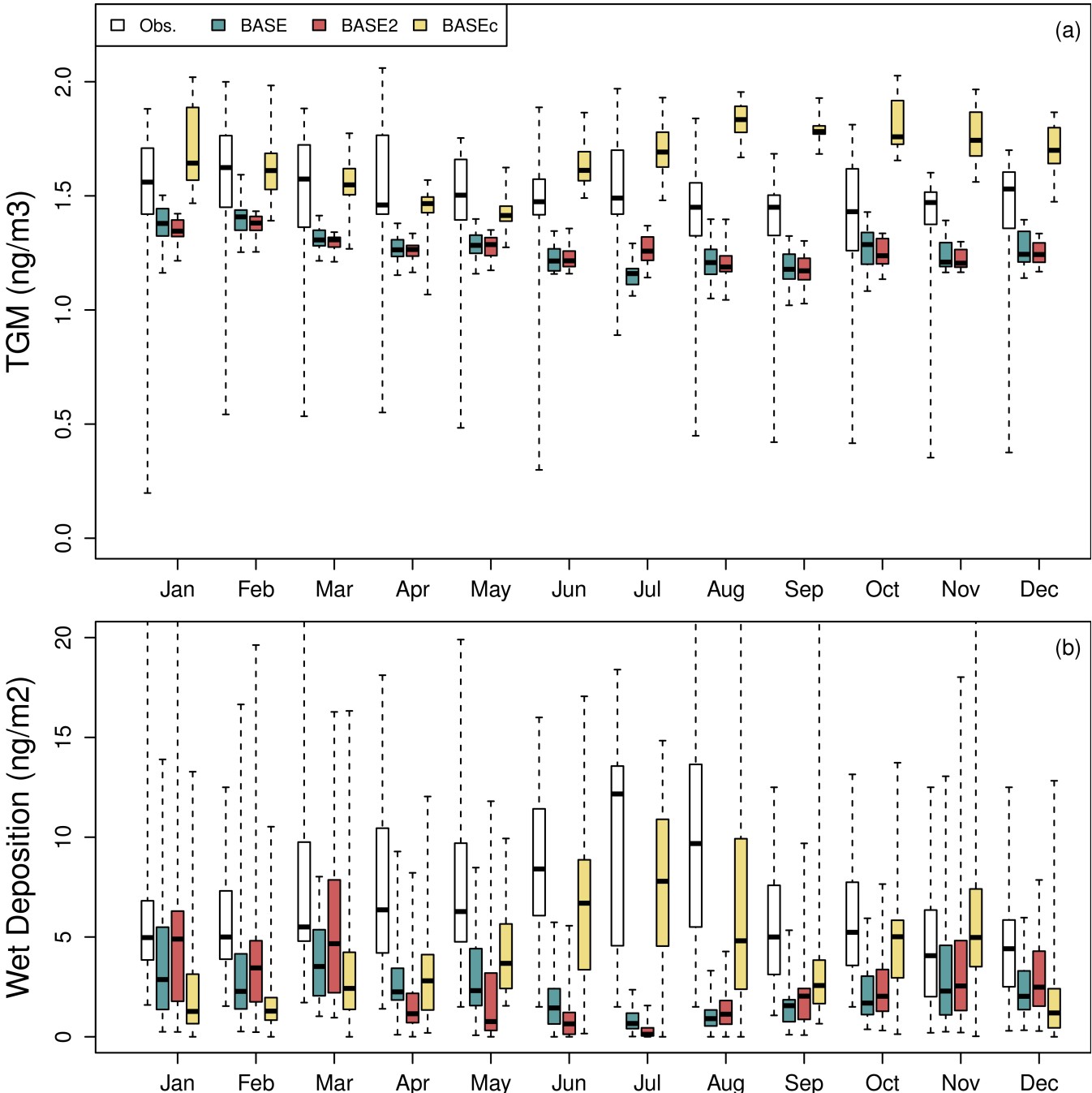

**Figure 6.** Monthly distribution of observed and modelled values for BASE, BASE2 and BASE$_C$ experiments at measurement stations. Upper panel (a) for TGM concentrations and lower panel (b) for wet deposition fluxes.

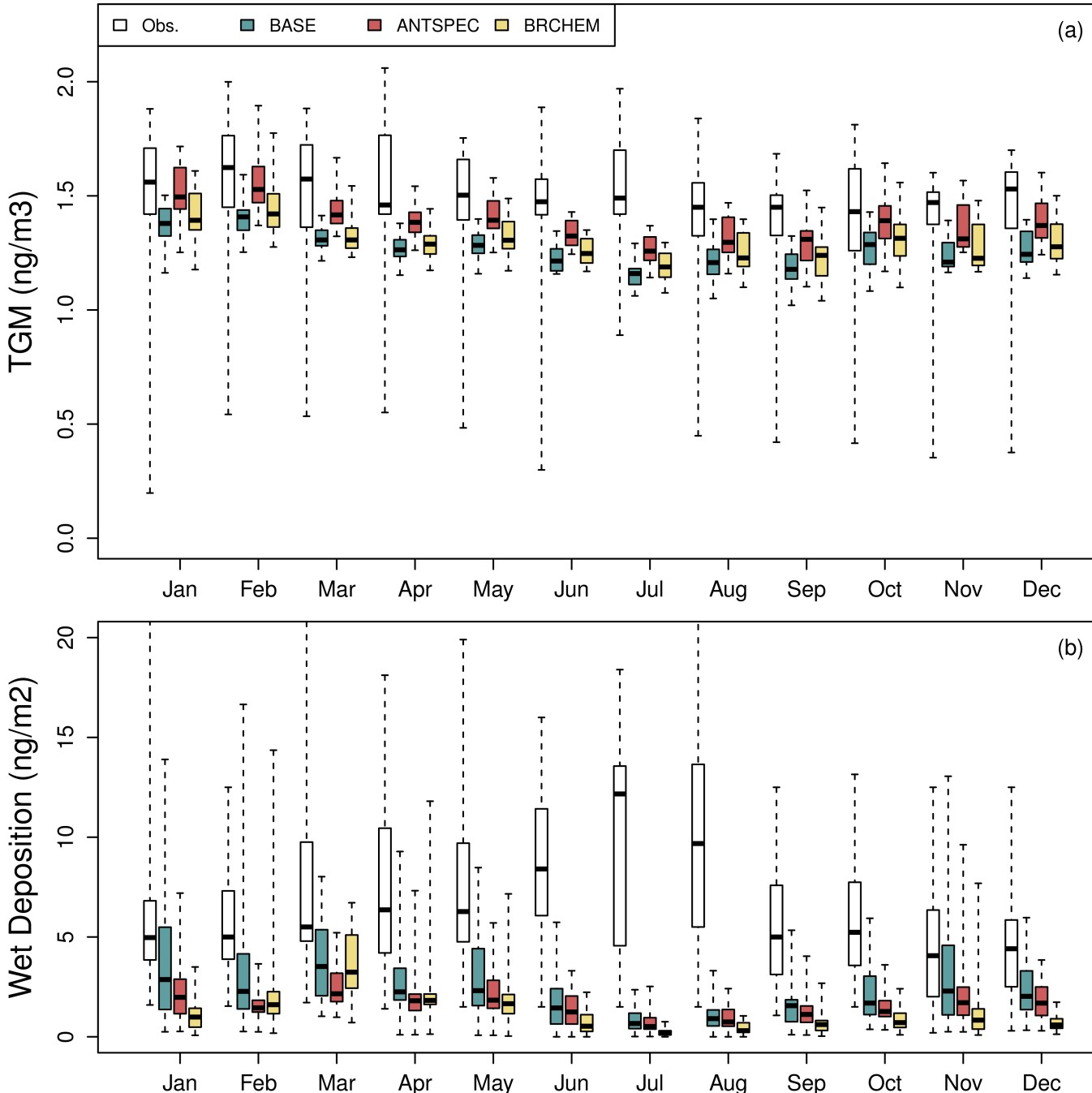

**Figure 7.** Monthly distribution of observed and modelled values for BASE, ANTSPEC and BRCHEM1 experiments at measurement stations. Upper panel (a) for TGM concentrations and lower panel (b) for wet deposition fluxes.

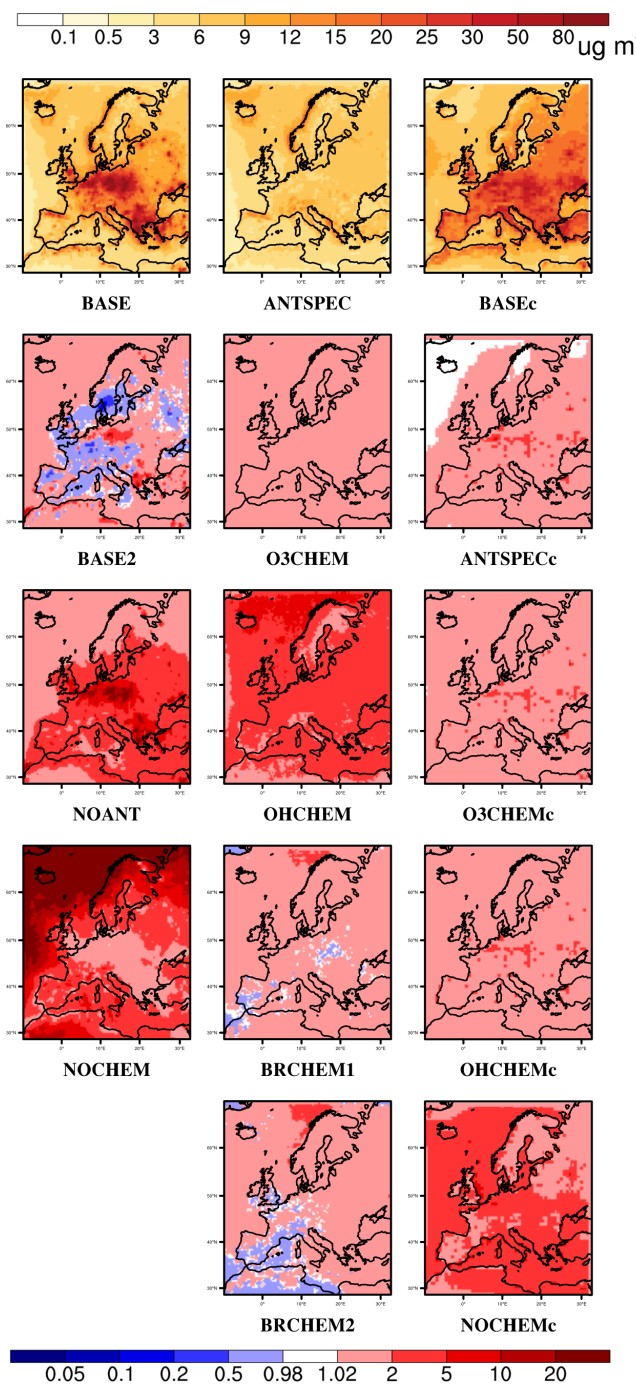

**Figure 8.** The total Hg deposition and the ratio of deposition in the sensitivity runs compared to BASE, ANTSPEC and BASE$_C$. The Hg total deposition is shown in the first row for BASE, ANTSPEC and BASE$_C$ experiments (upper color label), while the ratio (lower color label) of these with relative sensitivity runs is reported over the relative column: BASE2, NOANT and NOCHEM ratio (left column) are referred to BASE experiment, O3CHEM, OHCHEM, BRCHEM1 and BRCHEM2 (central column) ratio are referred to ANTSPEC experiment, ANTSPEC$_C$, O3CHEM$_C$, OHCHEM$_C$ and NOCHEM$_C$ (right row) ratio are referred to BASE$_C$ experiment.

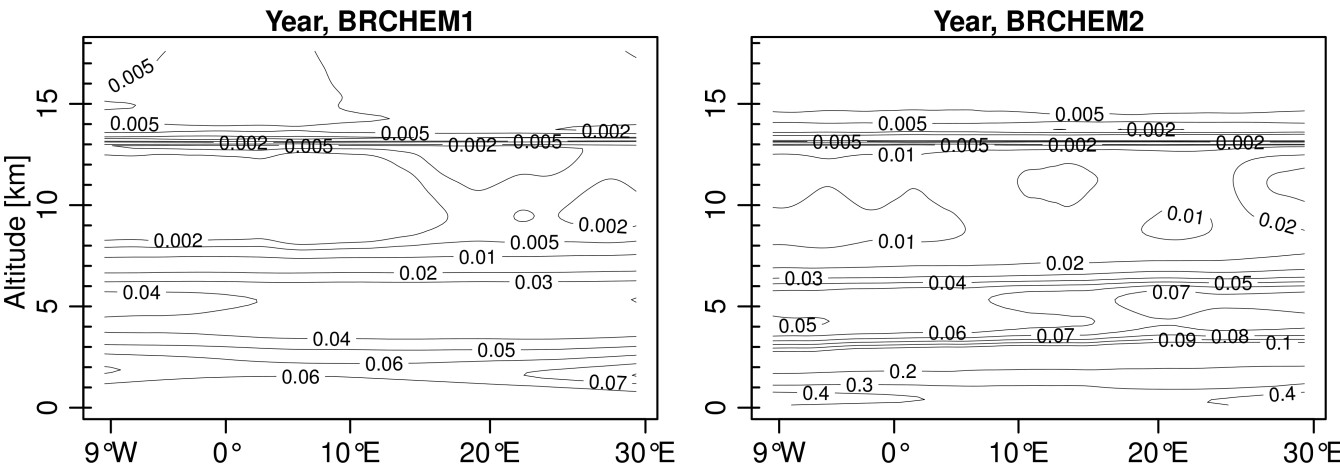

**Figure 9.** Vertical longitudinal profiles of Bromine annual mean concentrations (ppt) using p-TOMCAT concentrations (BRCHEM1 exeriment, left panel) and GEOSCHEM concentrations (BRCHEM2 exeriment, right panel).

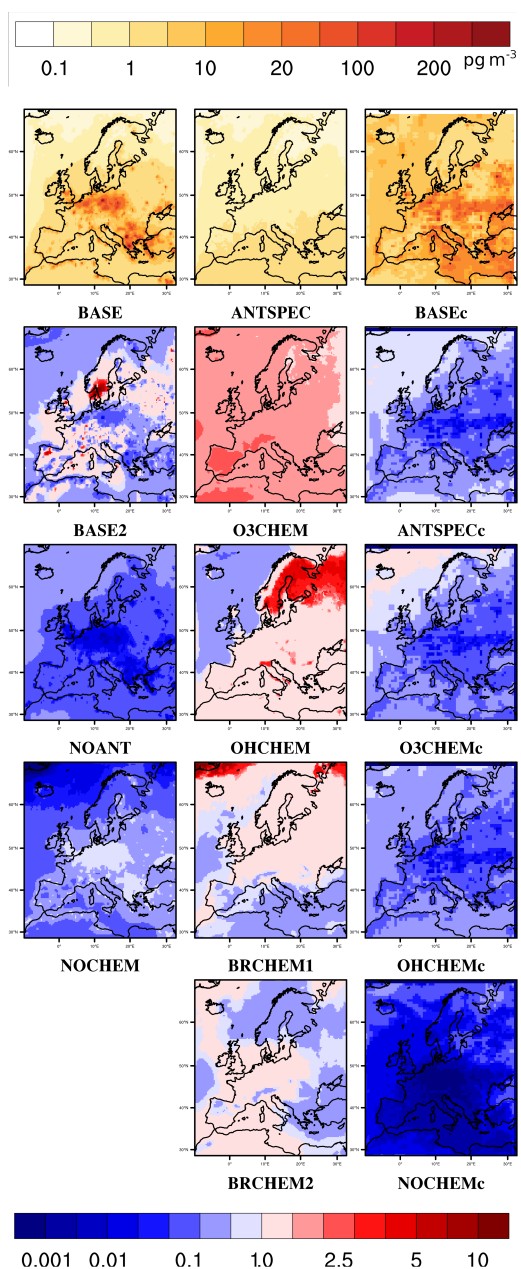

**Figure 10.** The annual RGM mean concentrations at ground level and the ratio of RGM in the sensitivity runs compared to BASE, ANTSPEC and BASE$_C$. The mean RGM concentrations are shown in the first row for BASE, ANTSPEC and BASE$_C$ experiments (upper colour label), while the ratio (lower colour label) of these with the sensitivity runs is reported in the relative column: BASE2, NOANT and NOCHEM ratio (left column) are referred to the BASE experiment, O3CHEM, OHCHEM, BRCHEM1 and BRCHEM2 (central column) ratios refer to the ANTSPEC experiment, ANTSPEC$_C$, O3CHEM$_C$, OHCHEM$_C$ and NOCHEM$_C$ (right column) ratios refer to the BASE$_C$ experiment.

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
