# Peer review of "Sensitivity model study of regional mercury dispersion in the atmosphere"

_Atmospheric Chemistry and Physics, 2016_

## Short Comment (SC1) · 8 Aug 2016

The mechanism used here for Br-initiated oxidation of mercury (from Gencarelli et al (2015)) cites Goodsite et al (2004) for the rate constants for HgBr reacting with Br and OH. These two rate constants are given as equal in Gencarelli et al. However, Goodsite et al did not report a rate constant for HgBr reaction with OH, so Gencarelli et al (2015) assumed that these two rate constants are equal.

If a model is going to treat these two rate constants as equal, it seems reasonable to include the reactions of HgBr with NO2, HOO, ClO, and BrO with the same rate constant, as was suggested by Dibble et al (2012).

References

[Figure]

Dibble, T. S., Zelie, M. J., and Mao, H. Thermodynamics of reactions of HgCl and HgBr radicals with atmospherically abundant free radicals. Atmos. Chem. Phys. 12, 10271-10279, 2012.

Gencarelli, C. N., De Simone, F., Hedgecock, I. M., Sprovieri, F., Yang, X., and Pirrone, N. European and Mediterranean mercury modelling: Local and long-range contributions to the deposition flux. Atmos. Env. 117, 162-168, 2015.

Goodsite, M. E., Plane, J. M. C., and Skov, H. A Theoretical Study of the Oxidation of Hg0 to HgBr2 in the Troposphere. Environ. Sci. Technol., 38, 1772-1776, 2004.
* * *

---

## Author Comment (AC1) · 24 Aug 2016

Dear Prof. Dibble,

thank you for your comment on our paper, and our apologies for the tardy reply. The HgBr + OH rate constant is taken from the assumptions made in Holmes et al. 2010 (Global atmospheric model for mercury including oxidation by bromine atoms, Atmos. Chem. Phys., 10, 12037–12057), thank you for pointing that out, we will correct the reference if and when the article is accepted.

We have actually added the reactions suggested in your 2012 article in our box model (AMCOTS, Sprovieri et al. 2010, An investigation of the origins of reactive gaseous mercury in the Mediterranean marine boundary layer, Atmos. Chem. Phys., 10, 3985–3997) assuming that the radical/compound + HgBr reaction is barrierless. We still fail

to reproduce the measured RGM values in the Mediterranean, due to the rapid rate of HgBr dissociation. We use the case of our studies in the Mediterranean because of the anticyclonic conditions which render the box model at least partly valid due to the atmospheric stability.

The Abstract from Wang et al. 2014 (Enhanced production of oxidised mercury over the tropical Pacific Ocean: a key missing oxidation pathway, Atmos. Chem. Phys., 14, 1323-1335) states: "We conclude that the key pathway that significantly enhances atmospheric mercury oxidation and deposition to the tropical oceans is missing from the current understanding of atmospheric mercury oxidation", and we would agree.

The stability of the HgBr radical is fundamental, and certainly in polar regions and one assumes, the free troposphere, where its thermal stability is less of an issue then it is very important in the procuction of Hg(II).

As has been suggested a number of times (for example, Subir et al. 2012, A review of the sources of uncertainties in atmospheric mercury modeling II. Mercury surface and heterogeneous chemistry – A missing link, Atmos. Environ., 46, 1-10), perhaps homogeneous oxidation of Hg in the atmosphere is only part of the equation.

The primary aim of the article is to look at the impact of Hg emission speciation and emissions vertical distribution. We are currently working on an extended chemical mechanism for the Bromine initiated Hg oxidation chemistry option, we hope to have finished the update soon, and should be running simulations this winter, with the hope of publishing the results earloy next year.

Best regards, Christian and Ian
* * *

---

## Referee Comment (RC1) · Anonymous Referee #1 · 7 Sep 2016

The manuscript reports the model results of a number of carefully crafted sensitivity simulations under different scenarios of mercury emission inventory (AMAP and EDGAR) and chemical oxidation ($O_3$, OH and Br) of elemental mercury in air, using WRF/Chem-Hg and CMAQ-Hg. The model results and data interpretation are presented in a organized fashion; and the conclusions are useful for better understanding of the chemical transport of mercury at regional scale. There are a few minor points that can be explained in more detail. I recommend the manuscript be accepted for publication after clarifying the points or providing the discussion of the following:

P4L27. The authors claim that the deposition parameterization does not have an effect on the ratio of simulate dry to wet deposition. This is somewhat surprising and should be clarified.

[Figure]

Figure 5. The authors rank the ratios from high to low. Discussion should be provided regarding what causes the spatial difference should be provided.

It appears that (1) the variability of simulated concentration does not match the variability of observed concentration, and (2) the simulated wet deposition grossly underestimate the observed values. (Figures 6 and 7). Discussion regarding the reasons should be provided.

It is somewhat disappointing that the authors paid little attention to the simulated concentration of gaseous oxidized mercury, considering the experiment on the oxidation mechanism of gaseous elemental mercury. Discussion around this topic is of scientific interest and should be provided.

After such an expansive modeling assessment, the authors may want to provide a synthesized conclusion regarding regional model configuration (chemistry, emission, etc.) for atmospheric modeling.

---

## Referee Comment (RC2) · Anonymous Referee #2 · 8 Sep 2016

This paper presents the results of model sensitivity studies of anthropogenic emissions and oxidation chemistry on atmospheric concentrations and deposition of mercury over Europe. The model results of each study are evaluated against observations.

General comments:

- I recommend the paper for publication after the revisions suggested here, which are largely about adding more explanation of the sensitivity studies and results, including more connection of the results to the current state of knowledge in the field, and highlighting implications of the results.

- In general, the discussion can be hard to follow at times since only some of the results are presented in the figures but results from all sensitivity studies are discussed.

- Every option for emissions inventories, chemistry schemes, etc. are presented as equally valid but it would be helpful to know what discussion has already been performed of these inventories/chemical mechanisms etc. to indicate the strengths and weaknesses of each and how this paper's results fit in. I think it is misleading to present Br and OH/O3 oxidation mechanisms on equal footing, for example, but then the paper shows that Br oxidation alone best represents TGM results which adds further support to what previous studies have suggested.

- I think the paper would benefit from more clearly outlining the purpose of each of the sensitivity studies. It would also be beneficial to have the explanation of most sensitivity studies be more centralized – since some of the sensitivity studies are untrue hypotheticals (e.g., NOCHEM and NOANT) while others are testing different hypothesized physical processes (e.g., BRCHEM1 vs. BRCHEM2), but both types are valuable. There is some of this scattered throughout the results section, but I think it can be more explicit; e.g., 'the ANTSPEC experiment, which assumes all anthropogenic emissions are as GEM, would represent a lower bound on deposition from local anthropogenic sources and an upper bound on long-range transport of anthropogenic emissions because GEM has a much longer lifetime against deposition than RGM/PBM' or something to that effect.

- Some background information is missing – e.g., the lifetime of GEM vs. RGM/PBM is not explained in the paper, or the sinks of GEM vs. RGM/PBM (dry deposition vs. wet deposition affinities), making the results from the ANTSPEC simulations and others less easily understood. Some aspects of atmospheric mercury are missing from the discussion (e.g., reduction of Hg(II), uncertainty in present oxidized Hg measurement capabilities that would be relevant for the 2nd to last sentence of the paper).

Specific comments:

Section 2.1: There are a lot of details given here about the models but it is not clear how the model representations of Hg are different or how their differences would affect Hg

results. It would be helpful to rewrite this a bit and make it more explicit whether/how the differences would affect Hg (e.g., if CMAQ is offline while WRF is online meteorology, the transport of Hg will be different to some degree between the models.)

Page 4 line 2: it would be helpful to explain briefly here whether there are multiple heights in the EDGAR inventory, as it is written it is unclear what the difference is and this comes up later in the results section.

Section 2.3: It is unclear from the way the paper is written what the base model oxidation chemistry schemes include –is it OH, O3 and Br oxidation at the same time in both CMAQ and WRF-Chem Hg? This should be made clear.

Page 4 Lines 23-25: some of this would be helpful to mention in the model description section 2.1.

Table 3: add a total deposition column, just makes it easier to follow

Figure 3: It is hard to compare the results between CMAQ and WRF-Chem Hg because the same model setups do not line up. I think it is worth reordering the individual panels to lining up the ANTSPEC and ANTSPEC_C vertically, or NOCHEM with NOCHEM_C below it, etc. to make more obvious the comparisons made in Section 3 of the paper.

Figure 4: This is total dry deposition of Hg(II)+Hg(0), correct? Just making sure since wet deposition is of course just Hg(II). I think this is worth mentioning as it also helps explain the changes in dry vs. wet deposition seen in the sensitivity studies.

Figure 5a: there are two legend labels with "BASE". I assume the green square is BASE2? Also, I would explain that the order of the sites is by the magnitude of the mod/obs ratio of the BASE simulation as it is not immediately obvious.

Page 5 Lines 20-23. It is counterintuitive to me how an overestimate relative to observations for the ANTSPEC sensitivity study where all anthropogenic emissions of RGM & PBM are as GEM means long-range transport is less important than regional emissions. I think more explanation would be helpful – so when PBM and RGM are

emitted normally as in the BASE simulation, the model is no longer overestimating the observations because the Hg(II) is deposited fast enough before it reaches the CZ03 site? I suppose part of this is defining what is local vs. "regional" vs. long-range more clearly.

Page 5 last paragraph (beginning line 29): do you have any hypotheses why TGM at ES08 is so low relative to the models?

Page 6 Lines 1-2: The discussion of OHCHEM and NOANT experiments comes as a surprise as it is not in Figures 5, 6, or 7; more connection to the rest of the section is needed.

Page 7 Lines 15-16: it is also not immediately obvious why dry deposition decreases so much more than wet deposition, since GEM is not wet deposited. Is it because even though there is higher GEM in the ANTSPEC experiment, it is dry deposited so much more slowly than the RGM/PBM species, or is it something about how RGM and PBM contribute different amounts to wet vs. dry dep of the oxidized species? Setting up the background on this in the intro or methods would be helpful.

Page 7 lines 21-23: given that a no anthropogenic emissions scenario is currently untrue, I think something more insightful can be said about the results – e.g., something about how anthro. emissions contribute to 2/3 of total deposition (not counting the fact that "natural" emissions from soil/ocean as they are tuned in models also implicitly are impacted by legacy anthropogenic sources), or how a hypothetical policy scenario of shutting off emissions could have huge local benefits?

Section 3.3 paragraph 2 (pg 7 lines 30-32): I would emphasize that this shows a significant proportion (exact percent varying on the model) of total Hg deposition to ecosystems is coming from the oxidation of GEM which can be transported from far distances as opposed to the Hg(II) locally emitted. This is an interesting result with policy implications and could be highlighted more.

Page 8 lines 3-4: "A number of studies have shown the importance of O3, and the OH radical, and also reactive halogen compounds….": I understand that there have been review papers discussing the intricacies of this and you don't want to repeat that here, but it is overly simple to group all three oxidants together and not mention that studies have found that the homogeneous gas-phase oxidation of Hg(0) by O3 and OH are thermodynamically and/or kinetically impossible (e.g., Hynes et al. (2009), Goodsite et al. (2004), Calvert and Lindberg, 2005). I think it is still interesting to compare the results from the three species, but it needs to be introduced with a bit more nuance. Moreover, as Theodore Dibble posted in his comment, there are additional HgBr+X second-step oxidation reactions that can greatly increase the total Hg(II) production and deposition through the Br-initiated pathway. Somewhere in the paper there should be a discussion of how this would affect the results presented – e.g., Hg deposition in the BRCHEM1 and 2 sensitivity simulations would be increased and TGM would be decreased.

Page 9 Lines 1-8: it is hard to understand the differences in the two Br concentration fields from the description given – would it be possible to show (perhaps in the supplemental) a difference plot of the Br concentrations over the Europe domain? (e.g., zonal mean latitude on x-axis vs. altitude on y-axis or something like that). Most GEM oxidation is not occurring in the PBL but in the free and upper troposphere (because of Br distribution and the temperature dependence of the oxidation reactions), so it is not surprising that the huge differences in Br in the PBL between the two Br fields doesn't impact on Hg(II) deposition; I am more interested in the differences at higher altitudes.

Page 9 Lines 13-16: I think it is essential to connect this to available observations of Br. The Shah et al. study tripled bromine concentrations of the GEOS-Chem Parrella et al. 2012 model which was consistent with observations of BrO during the NOMADSS field campaign (Gratz et al., 2015). Parrella et al. (2012) showed previously that BrO was underestimated in GEOS-Chem by 30% in the global mean against satellite observations. So it is not just a model exercise but shows that higher Br in GEOS-Chem (closer

to reality but not quite there) is also closer to reality for Hg model results. But in addition – since anthropogenic Hg(II) emissions have been turned to GEM in reality there would be more deposition with just Hg+Br oxidation and anthropogenic emissions of RGM/PBM turned on. This makes me curious about the Hg(II) reduction mechanisms in CMAQ vs. WRF-Chem Hg – if this was included, was it treated the same in all sensitivity simulations? I think reduction needs to be discussed somewhere in the paper.

additional references cited here:

Calvert, J. G., and S. E. Lindberg (2005), Mechanisms of mercury removal by O-3 and OH in the atmosphere, Atmospheric Environment, 39(18), 3355-3367, doi:10.1016/j.atmosenv.2005.01.055.

Goodsite, M., J. Plane, and H. Skov (2004), A theoretical study of the oxidation of Hg-0 to HgBr2 in the troposphere, Environmental Science & Technology, 38(6), 1772-1776, doi:10.1021/es034680s.

Gratz, L. E., Ambrose, J. L., Jaffe, D. A., Shah, V., Jaeglé, L., Stutz, J., Festa, J., Spolaor, M., Tsai, C., Selin, N. E., Song, S., Zhou, X., Weinheimer, A. J., Knapp, D. J., Montzka, D. D., Flocke, F. M., Campos, T. L., Apel, E., Hornbrook, R., Blake, N. J., Hall, S., Tyndall, G. S., Reeves, M., Stechman, D., and Stell, M.: Oxidation of mercury by bromine in the subtropical Pacific free troposphere, Geophys. Res. Lett., 42, 10494–10502, doi:10.1002/2015GL066645, 2015a.

Hynes, A. J., D. L. Donohoue, M. E. Goodsite, and I. M. Hedgecock (2009), Our Current Understanding of Major Chemical and Physical Processes Affecting Mercury Dynamics in the Atmosphere and At the Air-Water/Terrestrial Interfaces, in Mercury Fate and Transport in the Global Atmosphere, edited by N. Pirrone and R. Mason, pp. 427-457, Springer Science+Business Media, LLC, doi:10.1007/978-0-387-93958-2_14.

---

## Author Response (AR1)

The manuscript reports the model results of a number of carefully crafted sensitivity simulations under different scenarios of mercury emission inventory (AMAP and EDGAR) and chemical oxidation (O3, OH and Br) of elemental mercury in air, using WRF/Chem-Hg and CMAQ-Hg. The model results and data interpretation are presented in a organized fashion; and the conclusions are useful for better understanding of the chemical transport of mercury at regional scale. There are a few minor points that can be explained in more detail. I recommend the manuscript be accepted for publication after clarifying the points or providing the discussion of the following:

P4L27. The authors claim that the deposition parametrisation does not have an effect on the ratio of simulate dry to wet deposition. This is somewhat surprising and should be clarified.

*We think the referee may have misread the sentence,*
*"The differences in deposition parametrisations does have an effect on the ratio of dry to wet Hg deposition however."*
*We purposefully used "does have" rather than "has" in order to emphasize the fact that the dry to wet deposition ratio changes, as is clearly stated in the sentence that follows,*
*"While dry and wet deposition are almost equal in the WRF simulations (wet 49%, dry 51%), the dry deposition in CMAQ is more than twice the wet (69% dry and 31% wet), see Table 3, Fig. 3 and Fig. 4 for details."*

*We have therefore left the text unchanged.*

Figure 5. The authors rank the ratios from high to low. Discussion should be provided regarding what causes the spatial difference should be provided.

*This is an interesting suggestion. Unfortunately it was not possible to identify obvious spatial patterns, although in the stations located around the Baltic Sea a general overestimation of WD measurements by the model is noted.*
*Generally for GEM atmospheric concentrations there is a general underestimation in the WRF model simulations and an overestimation in CMAQ model simulations.*
*For wet deposition values the CMAQ model tends overestimate the observations, especially in Scandinavia, England and at Longobucco. On the other hand the WRF model has different characteristics: in Scandinavia the observations are always overestimated when compared to the rest of the domain, in the BASE2 experiment the greatest overestimation occurs while in the ANTSPEC experiment there is a general underestimation almost everywhere (given the lack of RGM emissions it is not surprising that the deposition is lower in this experiment).*

*These comments were added to section 3.1.*
*according*

It appears that (1) the variability of simulated concentration does not match the variability of observed concentration, and (2) the simulated wet deposition grossly underestimate the observed values. (Figures 6 and 7). Discussion regarding the reasons should be provided.

*(1) It is normal that the ratio is not equal between measurements and model values, otherwise the ratio should always be 1 (perfect agreement). But a fairly accurate agreement consists in having*

*the ratio within the range of uncertainty in the literature (dashed lines in figure 5: 30% for the GEM air concentrations, figure 5a, and factor 5 for wet deposition, figure 5b)*
*(2) Regarding the wet deposition in the estimation of these fluxes many more factors are involved (e.g. estimation of rainfall, coalescence efficiency), which increase the inaccuracies between observed and modeled values. In fact the literature range of the uncertainty is much larger.*

*With these suggestions the comments to Figure 5 have been expanded*

It is somewhat disappointing that the authors paid little attention to the simulated concentration of gaseous oxidized mercury, considering the experiment on the oxidation mechanism of gaseous elemental mercury. Discussion around this topic is of scientific interest and should be provided.

*We initially planned to only discuss the model results which in part can be compared with the observations. Considering that we do not have enough RGM measures to harmonize the discussion, we avoided including these results. However these results have been added in section 3.3 and the maps in figure 10 now show the average concentrations of modelled RGM for the main experiments (BASE, ANTSPEC and BASEc) and the ratio with respect to the sensitivity runs.*

After such an expansive modelling assessment, the authors may want to provide a synthesized conclusion regarding regional model configuration (chemistry, emission, etc.) for atmospheric modelling.

*A more extensive description of the models and the differences between them was added in Section 2.1, as suggested also by Referee #2.*

Interactive comment on "Sensitivity model study
of regional mercury dispersion in the atmosphere"
by Christian N. Gencarelli et al.
Anonymous Referee #2

This paper presents the results of model sensitivity studies of anthropogenic emissions and oxidation chemistry on atmospheric concentrations and deposition of mercury over Europe. The model results of each study are evaluated against observations.

General comments:
- I recommend the paper for publication after the revisions suggested here, which are largely about adding more explanation of the sensitivity studies and results, including more connection of the results to the current state of knowledge in the field, and highlighting implications of the results.

- In general, the discussion can be hard to follow at times since only some of the results are presented in the figures but results from all sensitivity studies are discussed.

*In the different sections only the figures regarding the sensitivity tests that were useful to compare with observations are shown. Otherwise the article would have been overly long.*

- Every option for emissions inventories, chemistry schemes, etc. are presented as equally valid but it would be helpful to know what discussion has already been performed of these inventories/chemical mechanisms etc. to indicate the strengths and weaknesses of each and how this paper's results fit in. I think it is misleading to present Br and OH/O3 oxidation mechanisms on equal footing, for example, but then the paper shows that Br oxidation alone best represents TGM results which adds further support to what previous studies have suggested.

*Text added: "A summary of the simulations performed is shown in Table 1. Some of these tests have been studied for other regions (e.g. Travnikov et al. (2016) and Bieser et al. (2016)) while many other studies have investigated Hg oxidation by Ozone or Br (Hynes et al., 2009; Subir et al., 2011, 2012; Weiss-Penzias et al., 2014)."*
*So in this paper we are not trying to make a direct comparison between the two mechanisms, but simply an analysis of how much the atmospheric Hg cycle may change considering a single Hg oxidant. The above text was added in section 2.3.*

- I think the paper would benefit from more clearly outlining the purpose of each of the sensitivity studies. It would also be beneficial to have the explanation of most sensitivity studies be more centralized – since some of the sensitivity studies are untrue hypotheticals (e.g., NOCHEM and NOANT) while others are testing different hypothesized physical processes (e.g., BRCHEM1 vs. BRCHEM2), but both types are valuable. There is some of this scattered throughout the results section, but I think it can be more explicit; e.g., 'the ANTSPEC experiment, which assumes all anthropogenic emissions are as GEM, would represent a lower bound on deposition from local anthropogenic sources and an upper bound on long-range transport of anthropogenic emissions because GEM has a much longer lifetime against deposition than RGM/PBM' or something to that effect.

*Very useful suggestions. Section 2.3 was expanded adding this information (bullets of different experiments).*

- Some background information is missing – e.g., the lifetime of GEM vs. RGM/PBM is not explained in the paper, or the sinks of GEM vs. RGM/PBM (dry deposition vs. wet deposition

affinities), making the results from the ANTSPEC simulations and others less easily understood. Some aspects of atmospheric mercury are missing from the discussion (e.g., reduction of Hg(II), uncertainty in present oxidized Hg measurement capabilities that would be relevant for the 2nd to last sentence of the paper).

*This information has been added in the Introduction*

Specific comments:
Section 2.1: There are a lot of details given here about the models but it is not clear how the model representations of Hg are different or how their differences would affect Hg results. It would be helpful to rewrite this a bit and make it more explicit whether/how the differences would affect Hg (e.g., if CMAQ is offline while WRF is online meteorology, the transport of Hg will be different to some degree between the models.)

*A more extensive models description was added in Section 2.1*

Page 4 line 2: it would be helpful to explain briefly here whether there are multiple heights in the EDGAR inventory, as it is written it is unclear what the difference is and this comes up later in the results section.

*Text added: "They also have different emission height distributions: AMAP uses three height classes (0-50, 50-150 and above 150\,m) whereas EDGAR ranges into six classes (distributed between 0 and 800 metres, listed according with SNAP (Selected Nomenclature for Air Pollution) categories as used in De Simone et al. (2016)). The differences in the geographical distributions are shown in Fig. 2." The EDGAR anthropogenic ranges in six classes, distributed between 0 and 800 m, listed according with SNAP categories.*
*This was added in Section 2.2.*

Section 2.3: It is unclear from the way the paper is written what the base model oxidation chemistry schemes include –is it OH, O3 and Br oxidation at the same time in both CMAQ and WRF-Chem Hg? This should be made clear.

*The sentence was modified to explain this point: "The BASE experiment refers to the model in the standard configuration, with AMAP anthropogenic emissions distributed according with the inventory guidelines." has been changed to "The BASE experiment refers to the model in the standard configuration, with AMAP anthropogenic emissions and Hg oxidation driven only by O3 and OH for WRF/Chem-Hg and by O3, OH, H2O2 and Cl2 for CMAQ-Hg, as described in section 2.1".*

Page 4 Lines 23-25: some of this would be helpful to mention in the model description section 2.1.

*Sections 2.1 and 3 were modified following this suggestion.*

Table 3: add a total deposition column, just makes it easier to follow

*Table 3 was modified following this suggestion.*

Figure 3: It is hard to compare the results between CMAQ and WRF-Chem Hg because the same model set ups do not line up. I think it is worth reordering the individual panels to lining up the ANTSPEC and ANTSPEC_C vertically, or NOCHEM with NOCHEM_C  below it, etc. to make more obvious the comparisons made in Section 3 of the paper.

*We are not sure that it is convenient to change the order of the figures. The maps should be read horizontally, with the first column (BASE, ANTSPE and BASEc) as a reference for the other maps.*

Figure 4: This is total dry deposition of Hg(II)+Hg(0), correct? Just making sure since wet deposition is of course just Hg(II). I think this is worth mentioning as it also helps explain the changes in dry vs. wet deposition seen in the sensitivity studies.

*No, it is just wet deposition flux, not the sum. Only the total dry (figure 3) and total wet (figure 4) deposition contributions are reported in the figures, to underline the results of tests. Total dry and total wet deposition are obtained as the sum of gaseous species and particulate deposition.*

Figure 5a: there are two legend labels with "BASE". I assume the green square is BASE2? Also, I would explain that the order of the sites is by the magnitude of the mod/obs ratio of the BASE simulation as it is not immediately obvious.

*The legend of figure 5 was modified following this suggestion.*

Page 5 Lines 20-23. It is counterintuitive to me how an overestimate relative to observations for the ANTSPEC sensitivity study where all anthropogenic emissions of RGM & PBM are as GEM means long-range transport is less important than regional emissions. I think more explanation would be helpful – so when PBM and RGM are emitted normally as in the BASE simulation, the model is no longer overestimating the observations because the Hg(II) is deposited fast enough before it reaches the CZ03 site? I suppose part of this is defining what is local vs. "regional" vs. long-range more clearly.

*First, the definition of Local and Long-Range follows that of Gencarelli et al., 2015: Local are  the sources of Hg inside the domain (anthropogenic emissions and evasion from the sea surface), Long-Range sources of Hg come the from boundary condition (external to domain). About the overestimate relative to observations for the ANTSPEC experiment, it concern also the GEM air concentrations (I was wrong not specify this), in fact the depositions in CZ03 station are lower in the ANTSPEC than the BASE experiment. Section 3.1 was modified following this suggestion.*

Page 5 last paragraph (beginning line 29): do you have any hypotheses why TGM at ES08 is so low relative to the models?

*From the modelling point of view of this study no clues have emerged to explain these low concentrations. The causes are probably attributable to the measurement technique or some unusual combination of local phenomena.*

Page 6 Lines 1-2: The discussion of OHCHEM and NOANT experiments comes as a surprise as it is not in Figures 5, 6, or 7; more connection to the rest of the section is needed.

*In order to include OHCHEM and NOANT  to the rest of discussion, the results of these experiment are reported in figure 5. It is not  surprising that this sensitivity test yields very low concentrations in comparison with total deposition and overestimation with air concentrations.*

Page 7 Lines 15-16: it is also not immediately obvious why dry deposition decreases so much more than wet deposition, since GEM is not wet deposited. Is it because even though there is higher GEMair  in the ANTSPEC experiment, it is dry deposited so much more slowly than the RGM/PBM species, or is it something about how RGM and PBM contribute different amounts to wet vs. dry dep of the oxidized species? Setting up the background on this in the intro or methods would be helpful.

*This explanation has been added to secion 3.2:*
*In the ANTSPEC experiment the deposition decreases in comparison to BASE, in particular the dry decreases more than the wet. % RGM and PBM deposit more rapidly than GEM and so deposit in proximity to their emission sources where the air concentrations are higher. Clearly dry deposition can occur at any time while wet deposition requires precipitation. With all \ce{Hg} emissions releases treated as GEM in ANTSPEC the dry deposition decreases most as a result of the lack of direct emissions of RGM and PBM.*

Page 7 lines 21-23: given that a no anthropogenic emissions scenario is currently untrue, I think something more insightful can be said about the results – e.g., something about how anthro. emissions contribute to 2/3 of total deposition (not counting the fact that "natural" emissions from soil/ocean as they are tuned in models also implicitly are impacted by legacy anthropogenic sources), or how a hypothetical policy scenario of shutting off emissions could have huge local benefits?

*The final part of section 3.2 was modified according to this suggestion and the results of Pacyna et al., 2016, that describes the results of modelling studies using GMOS project scenarios to assess Hg concentration and deposition fields, for present (2013) and future anthropogenic (2035) Hg emissions.*

Section 3.3 paragraph 2 (pg 7 lines 30-32): I would emphasize that this shows a significant proportion (exact percent varying on the model) of total Hg deposition to ecosystems is coming from the oxidation of GEM which can be transported from far distances as opposed to the Hg(II) locally emitted. This is an interesting result with policy implications and could be highlighted more.

*Section 3.3 and the conclusion have been modified according with this very useful suggestion.*

Page 8 lines 3-4: "A number of studies have shown the importance of O3, and the OH radical, and also reactive halogen compounds. . ..": I understand that there have been review papers discussing the intricacies of this and you don't want to repeat that here, but it is overly simple to group all three oxidants together and not mention that studies have found that the homogeneous gas-phase oxidation of Hg(0) by O3 and OH are thermodynamically and/or kinetically impossible (e.g., Hynes et al. (2009), Goodsite et al. (2004), Calvert and Lindberg, 2005). I think it is still interesting to compare the results from the three species, but it needs to be introduced with a bit more nuance. Moreover, as Theodore Dibble posted in his comment, there are additional HgBr+X second-step oxidation reactions that can greatly increase the total Hg(II) production and deposition through the Br-initiated pathway. Somewhere in the paper there should be a discussion of how this would affect the results presented – e.g., Hg deposition in the BRCHEM1 and 2 sensitivity simulations would be increased and TGM would be decreased.

*The section 3.3 was modified according with this suggestion.*

Page 9 Lines 1-8: it is hard to understand the differences in the two Br concentration fields from the description given – would it be possible to show (perhaps in the supplemental) a difference plot of the Br concentrations over the Europe domain? (e.g., zonal mean latitude on x-axis vs. altitude on y-axis or something like that). Most GEM oxidation is not occurring in the PBL but in the free and upper troposphere (because of Br distribution and the temperature dependence of the oxidation reactions), so it is not surprising that the huge differences in Br in the PBL between the two Br fields doesn't impact on Hg(II) deposition; I am more interested in the differences at higher altitudes.

*To supplement the information about Bromine concentrations the vertical longitudinal profiles of the annual mean concentrations were added in section 3.3. The main differences between the two inputs are in the first 3 km and in the range 12-15 km, as shown in the new version of section 3.3.*

Page 9 Lines 13-16: I think it is essential to connect this to available observations of Br. The Shah et al. study tripled bromine concentrations of the GEOS-Chem Parrella et al. 2012 model which was consistent with observations of BrO during the NOMADSS field campaign (Gratz et al., 2015). Parrella et al. (2012) showed previously that BrO was underestimated in GEOS-Chem by 30% in the global mean against satellite observations. So it is not just a model exercise but shows that higher Br in GEOS-Chem (closer to reality but not quite there) is also closer to reality for Hg model results. But in addition – since anthropogenic Hg(II) emissions have been turned to GEM in reality there would be more deposition with just Hg+Br oxidation and anthropogenic emissions of RGM/PBM turned on. This makes me curious about the Hg(II) reduction mechanisms in CMAQ vs. WRF-Chem Hg – if this was included, was it treated the same in all sensitivity simulations? I think reduction needs to be discussed somewhere in the paper.

*The last part of section 3.3 was modified in accordance with this useful suggestion.*

additional references cited here:

Calvert, J. G., and S. E. Lindberg (2005), Mechanisms of mercury removal by O3 and OH in the atmosphere, Atmospheric Environment, 39(18), 3355-3367, doi:10.1016/j.atmosenv.2005.01.055.

Goodsite, M., J. Plane, and H. Skov (2004), A theoretical study of the oxidation of Hg0 to HgBr2 in the troposphere, Environmental Science & Technology, 38(6), 1772-1776, doi:10.1021/es034680s.

Gratz, L. E., Ambrose, J. L., Jaffe, D. A., Shah, V., Jaeglé, L., Stutz, J., Festa, J., Spolaor, M., Tsai, C., Selin, N. E., Song, S., Zhou, X., Weinheimer, A. J., Knapp, D. J., Montzka, D. D., Flocke, F. M., Campos, T. L., Apel, E., Hornbrook, R., Blake, N. J., Hall, S., Tyndall, G. S., Reeves, M., Stechman, D., and Stell, M.: Oxidation of mercury by bromine in the subtropical Pacific free troposphere, Geophys. Res. Lett., 42, 10494–10502, doi:10.1002/2015GL066645, 2015a.

[revised manuscript text omitted]

* * *
[1]removed: experiemntal

[2]removed: (Grell et al., 2005) which includes

[3]removed: emissions(anthropogenic and natural)

[4]removed: see Gencarelli et al. (2014a)for further details regarding

[5]removed: parametrisations and the physics options employed.

[6]removed: (Jung et al., 2009; De Simone et al., 2014)

The second model used is CMAQ-Hg (version 5.0.1), based on CMAQ (Byun et al., 1999) and modified by Bullock and Brehme (2002) and Gbor et al. (2006) to include chemistry, transport and deposition of GEM, GOM and PBM. This model was compiled with the multi-pollutant version of the CBM5 photochemical mechanism [..[7]](Sarwar et al., 2008) (which includes Hg gaseous reactions with $O_3$, OH, $H_2O_2$ and $Cl_2$ as described by Lin and Tao (2003)) with th eEuler Backward Iterative solver and the AERO4 aerosol mechanism (Binkowski and Roselle, 2003). [..[8] ]The CMAQ-Hg model uses offline meteorological fields provided by the COSMO-CLM model (Rockel et al., 2008)[..[9] ], processed by the Meteorology-Chemistry Interface Processor (MCIP v3.6). The same MCIP to calculate the dry deposition velocities of GEM and GOM. During the offline simulations cloud processes, including cloud attenuation of photolysis rates, convective and non-convective mixing and scavenging by clouds, aqueous-phase chemistry, and wet deposition were calculated as described in Liu and Zhang (2013). The chemical IC/BC were taken from [..[10] ]the GLEMOS model (Travnikov et al., 2009). For further details on the models see Gencarelli et al. (2014a, 2015) for WRF/Chem-Hg and Bieser et al. (2014); Zhu et al. (2015) for CMAQ-Hg.

The main difference between the two models is in the feedback between chemical and meteorological dynamics: while in CMAQ the meteorological fields are provided as input (offline model), in WRF they are solved simultaneously with the chemistry, in the same time step (online model). Other major differences concern the parametrisations of some of the processes, for instance, GEM dry deposition, convective precipitation and GEM evasion from the sea surface (see Gencarelli et al. (2015) and Bieser et al. (2014) for details). Other differences result from the use of different BC sets and meteorological input.

Oxidation of Hg by bromine was implemented in some of the WRF experiments, using the off-line [..[11] ]Br fields obtained from the p-TOMCAT (Yang et al., 2005, 2010) and GEOSCHEM (Parrella et al., 2012) models.

**2.2 Modelled emissions**

In order to analyse the effects of anthropogenic emissions speciation, amount and vertical distribution, the input from the two recent global anthropogenic Hg emission inventories were interpolated over the model grids and used in the sensitivity simulations.

The AMAP/UNEP 2010 (hereafter AMAP) inventory is available at a spatial resolution of 0.5° by 0.5° (AMAP/UNEP, 2013b), while the EDGARv4.tox1 2008 (hereafter EDGAR) has a spatial resolution of 0.1° by 0.1° (Muntean et al., 2014). Over the modelling domain the inventories differ in both emission totals and speciation ratio GEM[..[12] ][..[13] ]:RGM:PBM as:

– $136.2\,\mathrm{Mg\,y^{-1}}$ with GEM:RGM:PBM ratio 65:28:7 for AMAP, and
* * *
[7]removed: with updated chlorine and toluene chemistry (cb05tump) with on-line photolysis and the aero6 aerosol module. (see (Bieser et al., 2014; Zhu et al., 2015) for further details)

[8]removed: CMAQ model is an off-line model , with meteorology fields obtained from

[9]removed: . The chemical initial and boundary conditions

[10]removed: base

[11]removed: Br

[12]removed: (Gaseous Elemental Mercury), RGM(Reactive Gaseous Mercury,

[13]removed: ), PBM (Particulate Bound Mercury)

– 123.8 Mg y$^{-1}$ with 60:32:8 for EDGAR.

They also have different emission height distributions: AMAP uses three height classes (0-50, 50-150 and above 150 m) whereas EDGAR [..[14] ]ranges into six classes (distributed between 0 and 800 metres, listed according with SNAP (Selected Nomenclature for Air Pollution) categories as used in De Simone et al. (2016)). The differences in the geographical distributions are shown in Fig. 2.

**2.3 Simulations performed**

Simulations were performed varying the emissions speciation, the atmospheric Hg oxidation mechanism, the bromine concentration field and the atmospheric process parametrisation. A total of 14 (9 with WRF and 5 with CMAQ) 12-month model simulations were conducted, as reported in Table 1, where experiments conducted using CMAQ are indicated by a $C$ subscript. The [..[15] ]specific scopes of every particular experiment as:

– BASE – base case test, used as reference experiment. It refers to the model in the standard configuration, with AMAP anthropogenic emissions [..[16] ]and Hg oxidation driven only by $O_3$ and OH for WRF/Chem-Hg and by $O_3$, OH, $H_2O_2$ and $Cl_2$ for CMAQ-Hg, as described in section 2.1.

– BASE2 – similar to BASE experiment, with the [..[17] ]only change of Hg anthropogenic emission used. In fact in this case EDGAR Hg emissions are used.

– NOANT – hypothetical scenario, where all anthropogenic emissions are turned off, in order to [..[18] ][..[19] ][..[20] ]highlight the influence of long-range transport on European areas. The same chemical mechanism of BASE experiment is used.

– NOCHEM - hypothetical scenario, where the chemical [..[21] ][..[22] ][..[23] ]reactions of Hg are turned off. In this way there are not conversion of GEM in RGM, that imply a different distribution of Hg deposition respect the BASE experiment.

– ANTSPEC – hypothetical experiment where all Hg emissions are treated as GEM. With this experiment RGM and PBM emission are turned off, and considering the different chemical and physical properties respect the GEM the
* * *
[14]removed: lists

[15]removed: BASE experiment

[16]removed: distributed according with the inventory guidelines. In the ANTSPEC experiment the RGM and PBM emissions are treated as GEM

[17]removed: aim to keep the total emissions constant and analyse the effects of emission speciation. In the NOANT experiment anthropogenic emissions were not included

[18]removed: estimate their effects, while to analise the effects of oxidation mechanisms the atmospheric

[19]removed: chemical reactions were excluded in the NOCHEM simulation. Simulations based on ANTSPEC were performed with a single

[20]removed: chemical reaction in the

[21]removed: mechanism, to assess the effects and feasibility of

[22]removed: reactions with

[23]removed: ,

[revised manuscript text omitted]